# Documentary-based climate reconstructions in the Czech Lands 1501–2020 CE and their European context

Rudolf Brázdil[1,2], Petr Dobrovolný[1,2], Jiří Mikšovský[2,3], Petr Pišoft[3], Miroslav Trnka[2,4], Martin Možný[5], and Jan Balek[2,4]

[1]Institute of Geography, Masaryk University, Brno, Czech Republic
[2]Global Change Research Institute, Czech Academy of Sciences, Brno, Czech Republic
[3]Department of Atmospheric Physics, Charles University, Prague, Czech Republic
[4]Department of Agrosystems and Bioclimatology, Mendel University in Brno, Brno, Czech Republic
[5]Czech Hydrometeorological Institute, Praha, Czech Republic

Correspondence: Rudolf Brázdil (brazdil@sci.muni.cz)

**Abstract.** Annual and seasonal temperature, precipitation and drought index (SPI – Standard Precipitation Index, SPEI – Standard Precipitation Evapotranspiration Index, Z-index, PDSI – Palmer Drought Severity Index) series covering the Czech Lands territory (now the Czech Republic) over 520 years (1501–2020 CE) reconstructed from documentary data combined with instrumental observations were analysed herein. The temperature series exhibits a statistically significant increasing trend, rising from ~1890 and particularly from the 1970s; 1991–2020 represents the warmest and driest 30-year period since 1501 CE. While the long-term precipitation total fluctuations (and derived SPI fluctuations) remain relatively stable with annual and decadal variabilities, past temperature increases are the key factor affecting recent increasing dryness in the SPEI, Z-index and PDSI series. The seasonal temperature series represent a broad European area, while the seasonal precipitation series show lower spatial correlations. A statistical attribution analysis conducted utilizing regression and wavelet techniques confirmed the influence of covariates related to volcanic activity (prompting temporary temperature decreases, especially during summer) and the North Atlantic Oscillation (influential in all seasons except summer) in the Czech climate reconstructions. Furthermore, components tied to multidecadal variabilities in the northern Atlantic and northern Pacific were identified in the temperature and precipitation series and in the drought indices, revealing notable shared oscillations, particularly at periods of approximately 70–100 years.

## 1 Introduction

Documentary evidence about weather and related phenomena is broadly used for different types of studies in historical climatology (e.g., Brázdil et al., 2005, 2010; White et al., 2018; Pfister and Wanner, 2021). To particularly describe temperature and precipitation patterns, temperature and precipitation indices were involved and used to create their long-term series, using most broadly 3- or 7-degree scales for the individual months (Pfister, 1992) but also other degree scales (see Nash et al., 2021 for overview). Many temperature/precipitation index series have been published in Europe, such as those for Switzerland (Pfister, 1988, 1999), the central part of European Russia (Lyakhov, 1992), central Europe (Glaser et al., 1999), the Low Countries (Shabalova and van Engelen, 2003; van Engelen et al., 2009), Germany (Glaser, 2008), the Mediterranean (Camuffo et al., 2010), Burgundian Low Countries (Camenisch, 2015), Gdansk, Poland (Filipiak et al., 2019), Buchlovice, Czech Lands (Brázdil et al., 2019), Sweden (Retsö and Söderberg, 2020), western and central Europe (Pfister and Wanner, 2021), and others.

However, it is difficult to compare series of temperature and precipitation indices with temperature or precipitation series expressed in standard units used for their measurements; it is in °C for temperature or in mm for precipitation. Although some attempts for quantitative expression of such series appeared earlier (e.g., Pfister and Brázdil, 1999; Brázdil and Kotyza, 2000; Glaser and Riemann, 2009), having a temporal overlap between series of indices and meteorological observations allowed us to apply a standard paleoclimatological approach for temperature/precipitation quantitative reconstructions, as was documented in the example of temperatures for Prague-Klementinum (Dobrovolný et al., 2009). Subsequently, a combination of a series of temperature indices for Germany, Switzerland and the Czech Lands with temperatures measured at 11 stations was used to quantitatively reconstruct monthly, seasonal and annual temperatures in central Europe for the past 500 years (Dobrovolný et al., 2010).

In addition to a series of indices interpreted from different documentary data, different (bio)physical series can also be used to reconstruct particular temperatures for any combination of months, for which analysed data are sensitive. It concerns many reconstructions based particularly on the dates of grain harvest beginnings (e.g., Wetter and Pfister, 2011; Pribyl et al., 2012), dates of grape harvest beginnings (e.g., Meier et al., 2007; Mariani et al., 2009; Maurer et al., 2009; Kiss et al., 2011; Daux et al., 2012; Molitor et al., 2016; Labbé et al., 2019), or dates of freezing of rivers, water channels and harbours (e.g., Tarand and Nordli, 2001; Leijonhufvud et al., 2008, 2010).

With respect to rich documentary evidence available in the Czech Lands (currently the Czech Republic), several series of temperature, precipitation and drought reconstructions starting from the beginning of the 16th century were created there. Despite the fact that the reconstructed temperature series of central Europe (Dobrovolný et al., 2010) is also representative of the Czech Lands, other temperature reconstructions are based on dates of winter wheat harvest (Možný et al., 2012) or grape harvest (Možný et al., 2016a). Based on a series of precipitation indices, Dobrovolný et al. (2015) reconstructed a series of seasonal and annual precipitation totals. Reconstructed temperature and precipitation series were subsequently used to compile a series of seasonal and annual drought indices (SPI, SPEI, Z-index, PDSI) of the Czech Lands (Brázdil et al., 2016). Moreover, Možný et al. (2016b) also used grape harvest dates to reconstruct a series of SPEI. There is hardly any other European country with so many documentary-based quantitative reconstructions as the Czech Lands. In addition to the reconstruction of Czech climatic characteristics and analysis of their inherent variability, attention has also been previously paid to identification of factors responsible for significant components imprinted in these series. In particular, the effects of external forcings and large-scale climate variability modes in the Czech long-term series of temperature, precipitation and drought indices were investigated by Mikšovský et al. (2014, 2019) and Brázdil et al. (2015), and added value from the use of multi-century reconstructions over observation-only data was highlighted.

The aim of the recent study is to present all Czech climate reconstructions extended on the 1501–2020 period together, to analyse their statistical features and their inter-relationships, effects of external forcings and large-scale climate variability modes, and finally to evaluate their spatiotemporal information ability with respect to other gridded climate reconstructions in Europe. Sect. 2 characterises shortly all available Czech climate reconstructions, series of variables pertaining to potential explanatory factors, and other European gridded reconstructions used for comparison. Methods used in this study are described in Sect. 3. The following Sect. 4 presents basic results oriented on inter-comparison of all Czech reconstructions, their statistical characteristics, the outcomes of attribution analysis and spatiotemporal comparison with gridded European climate reconstructions. The results obtained are discussed with respect to reconstruction uncertainties and the broader

context of the presented reconstructions in Sect. 5, followed by some conclusions in the last section.

**2 Data**

**2.1 Czech climate reconstructions**

The recent study uses the following climate reconstructions based on documentary data and instrumental observations related to the territory of the Czech Lands:

a) Temperature reconstructions

(i) Series of monthly, seasonal (DJF – winter, MAM – spring, JJA – summer, SON – autumn) and annual temperatures of central Europe (1500–2007 CE) based on temperature index series of Germany, Switzerland and the Czech Lands (1500–1854) and mean instrumental temperature series of 11 meteorological stations in central Europe (1760–2007) (Dobrovolný et al., 2010);

(ii) Series of March–June (MAMJ) temperatures of the Czech Lands (1501–2008 CE) derived from a series of winter wheat harvest dates (Možný et al., 2012);

(iii) Series of April–August (AMJJA) temperatures of the Czech Lands (1499–2015 CE) derived from a series of grape harvest dates (Možný et al., 2016a).

b) Precipitation reconstructions

(i) Series of seasonal and annual precipitation totals of the Czech Lands (1501–2010 CE) derived from documentary-based precipitation indices (1501–1854) and mean areal precipitation series of the Czech Republic (1804–2010) (Dobrovolný et al., 2015).

c) Drought reconstructions

(i) Series of seasonal and annual drought indices (Standard Precipitation Index SPI – McKee et al., 1993; Standard Precipitation Evapotranspiration Index SPEI – Vicente-Serrano et al., 2010; Z-index and Palmer Drought Severity Index PDSI – Palmer, 1965) of the Czech Lands for 1501–2015 CE (Brázdil et al., 2016), derived from central European temperature and Czech precipitation reconstructions (Dobrovolný et al., 2010, 2015);

(ii) Series of AMJJA SPEI indices of the Czech Lands (1499–2012 CE) derived from a series of grape harvest dates (Možný et al., 2016b).

For the purposes of this paper, all of the above series were taken from 1501 and extended until 2020 to cover the entire 1501–2020 CE period. For comparison with MAMJ and AMJJA temperatures by Možný et al. (2012, 2016a), MAMJ and AMJJA temperature series from temperature series of central Europe (Dobrovolný et al., 2010) were calculated. Similarly, the AMJJA SPEI series from Brázdil et al. (2016) was calculated for comparison with that of Možný et al. (2016b).

**2.2 European climate reconstructions**

To study the spatiotemporal representativeness of Czech climate reconstructions at the European scale, gridded European reconstructions are used:

a) Reconstruction of gridded ($0.5° \times 0.5°$) seasonal temperatures by Luterbacher et al. (2004) and Xoplaki et al. (2005) covering European land (25°W–40°E; 35°N–70°N) in the 1500–2002 period is spliced from temperature-sensitive natural and documentary proxy-based reconstructions before 1900 and instrumental measurements from Mitchell and Jones (2005) after that time (https://www.ncei.noaa.gov/pub/data/paleo/historical/europe-seasonal.txt, last access: 20 October 2021);

b) Reconstruction of seasonal precipitation by Pauling et al. (2006) includes gridded ($0.5° \times 0.5°$) totals for European land (30°W–40°E and 30°N–71°N) for the years 1500–1900 reconstructed from long instrumental precipitation series, documentary-based precipitation indices, and natural proxies (tree rings, ice cores, corals, and speleothems) combined with a

gridded reanalysis for 1901–2000 after Mitchell and Jones (2005)
(https://www.ncei.noaa.gov/access/paleo-search/study/6342, last access: 20 October 2021);
c) Reconstruction of summer self-calibrated (sc) PDSI for The Old World Drought Atlas
(OWDA) by Cook et al. (2015) includes gridded ($0.5° \times 0.5°$) data derived from tree ring
widths for the 0–2012 CE period (http://drought.memphis.edu/OWDA/Default.aspx, last
access: 20 October 2021).

Moreover, from gridded values of three mentioned gridded European reconstructions,
the mean series for the "central European window" with geographic coordinates 45°N–54°N
and 5°E–23°E was calculated and used for comparison with the Czech series applying 31-year
running correlation coefficients.

**2.3 External forcings and large-scale climate variability modes**
The following descriptors of external forcings and large-scale internal oscillatory climate
variability modes are used as explanatory variables in the attribution analysis:
a) Greenhouse gases radiative forcing (GHGRF)
Based on Meinshausen et al. (2011) annual data for the 1765–2020 period extended back to
1501 CE using the $CO_2$, $CH_4$ and $N_2O$ concentrations obtained from the online database of the
Institute for Atmospheric and Climate Science, ETH Zurich, and approximate formulas
provided in the IPCC report (IPCC, 2001, Table 6.2).
b) Solar activity (SOLAR)
Annual values of total solar irradiance by Lean (2018), extended to 2020 by data at
https://climexp.knmi.nl/data/itsi_ncdc_yearly.dat (last access: 20 October 2021).
c) Volcanic activity (VOLC)
Stratospheric volcanic aerosol optical depth (AOD) series in the 30°N–90°N latitudinal band,
adapted from reconstruction by Crowley and Unterman (2013), extended to 2020.
d) North Atlantic Oscillation (NAO)
Series by Luterbacher et al. (2001), available for 1659–2001 CE in monthly time steps and for
1500–1658 CE in seasonal time steps. Beyond 2001, NAO index values were calculated from
the standardized pressure difference between Iceland and the Azores using NCEP/NCAR
reanalysis data (Kalnay et al., 1996).
e) Atlantic Multidecadal Oscillation (AMO) and Pacific Decadal Oscillation (PDO)
Annual values of multidecadal temperature variations in the AMO and PDO regions by Mann
et al. (2009) were adopted for the 1501–2006 CE period and extended to 2020 by GISTEMP
(Hansen et al., 2010) areal temperature means for their respective northern Atlantic and
northern Pacific regions. To overcome problems with the strong mutual correlation of Mann
et al. (2009) AMO and PDO temperatures, their common component (designated
AMO+PDO) and difference (AMO-PDO) were used instead of the AMO and PDO series
themselves (following from predictor analysis presented in Mikšovský et al., 2019). The
common component (AMO+PDO) was further detrended by subtracting its component
correlated to greenhouse gases radiative forcing to more reliably separate signals related to
these two predictors.

**3 Methods**
Fluctuations in the Czech climate variables are expressed as annual and seasonal series
smoothed by a 30-year Gaussian filter and linear trends for which their significance was
calculated using a t test at the 0.05 significance level. For the entire 520-year series and the
most extreme 30-year periods (warmest and coldest; driest and wettest), corresponding box
plots (median, upper and lower quartile, maximum and minimum) are presented. Moreover,
using a t test, differences in the means of extreme 30-year periods compared to the mean of
the entire 520-year period were evaluated. For comparison of individual series, Pearson

correlation coefficients with their statistical significance according to t tests were also calculated. To compare temporal variability among different series, 31-year running correlation coefficients were applied. To demonstrate the representativeness of the Czech series at the European scale, maps of correlation coefficients were constructed.

To study cyclic components in Czech climatic series, a continuous wavelet transform, based on the Morlet mother wavelet, was applied (Torrence and Compo, 1998). The statistical significance of the wavelet coefficients was evaluated against an AR(1) process null hypothesis. Furthermore, cross-wavelet transform and wavelet coherence were applied to evaluate pairwise similarities in the time-frequency structure of individual time series. The GHGRF-correlated trend component was removed from all series before performing wavelet transform to reduce the effect of related long-term nonperiodic components on statistical significance estimates.

Multiple regression analysis was employed to quantify linear links between the explanatory variables and Czech climatic series. The results are presented through standardized regression coefficients, with statistical significance of the regression coefficients evaluated by moving-block bootstrapping (block size chosen to account for autocorrelations within the regression residuals – Politis and White, 2004; Bravo and Godfrey, 2012).

## 4 Results
## 4.1 Climate fluctuations in 1501–2020 CE
### 4.1.1 Series derived from temperature and precipitation indices

Fluctuations in annual temperature, precipitation and drought indices in the Czech Lands during the 1501–2020 period exhibit great interannual variability and prevailingly small nonsignificant linear trends (Fig. 1). Only for mean annual temperatures is the increasing trend statistically significant at the 0.05 significance level (0.11°C/100 years), when temperatures after preceding relatively stable fluctuations grew from *c*. 1890, and their increase was particularly enhanced starting in the 1970s. The last 30-year period of 1991–2020 experienced the highest temperatures in the whole series, while the coldest 30-year period was detected in 1829–1858 (Table 1). Very similar fluctuations characterise series of annual precipitation totals and SPI series derived from precipitation, experiencing no long-term trends. Both series agree in the wettest 1912–1941 period, while the driest 30-year episode occurred in the first three decades of the 18th century (with a small shift between the two variables). Three remaining series of drought indices show nonsignificant negative trends and agree in the driest 30-year interval of 1990–2019. For the wettest 30 years, the Z-index and PDSI agreed with the precipitation series during 1912–1941 (for the PDSI with a shift of one year), while in the SPEI it was already in the second half of the 16th century (1569–1598). The means of all selected 30-year extreme periods differ significantly from the means of the corresponding entire 520-year series.

Pearson correlation coefficients of annual temperatures with five other variables during the whole 1501–2020 period give statistically significant values between –0.27 with precipitation and –0.61 with SPEI. In terms of 31-year running correlations, they became to a greater extent statistically nonsignificant from the 19th century on, changing even signs of correlation from negative to positive approximately around 1900 (Fig. 2a). The close relationship of precipitation to drought indices with correlation coefficients from 0.59 with PDSI to 0.97 with SPI is well reflected in 31-year running correlations above the 0.05 significance level (Fig. 2b). The correlations among the four drought indices are the lowest between the SPI and PDSI (0.61) and the highest between the SPEI and Z-index (0.96). None of the 31-year running correlations between drought index series dropped below the significance level (Fig. 2c).

Similar features as in the case of annual series can also be detected in the corresponding seasonal series (Figs. 3–6). All temperature series agree in increasing 520-year linear trends (the highest in DJF 0.27°C/100 years and the lowest in SON 0.06°C/100 years), all statistically significant except SON, and in the warmest last three decades 1991–2020 (but DJF 1988–2017) (Table 1). A greater diversity appears in delimitation of the coldest 30-year periods: in the past three decades of the 16th century (DJF 1572–1601 and JJA 1569–1598), in the 18th century (SON 1757–1786) and in the 19th century (MAM 1832–1861). Seasonal series of precipitation totals and SPI indicate zero linear trends and a great variety in 30-year extreme periods. Distinct clustering in their occurrence appears only for the wettest intervals in DJF (1555–1584) and JJA (1568–1597), while the other two seasons have maxima at the end of the 19th century and the beginning of the 20th century (MAM 1885–1914) and during the first decades of the 20th century (SON 1910–1939). The driest 30-year intervals appeared during the entire 18th century and for only SON in the 17th century (1605–1634). Compared to the series of precipitation totals, the SPI series showed the different driest 30-year intervals in DJF (1680–1709) and partly shifted wettest 30 years in MAM (1894–1923). The three remaining seasonal drought indices experienced statistically nonsignificant negative 520-year linear trends. The driest last three decades 1991–2020 are typical for all seasonal Z-index and PDSI series (also for MAM and JJA SPEI). The wettest seasonal 30-year spans for the PDSI appear between 1912 and 1942 except MAM (1888–1917). Analogous intervals in the case of the Z-index overlap with those in the PDSI only partly (starting earlier), and in JJA, it occurs even during the 16th century in 1569–1598, in agreement with the SPEI. The second half of the 16th century also experienced the wettest 30 years for SPEI in DJF (1555–1584), while those for MAM nearly overlap with the Z-index and for SON with PDSI. In total, different from the Z-index and PDSI, the driest periods were in DJF (1680–1709) and in SON (1605–1634). Means of 30-year periods differed from the corresponding entire 520-year means only for the wettest and driest SPI in JJA and for the wettest SPI and SPEI in SON.

Relationships between seasonal temperature, precipitation and drought index series in the Czech Lands can be described using Pearson correlation coefficients in the entire 1501–2020 period, which are all statistically significant at the 0.05 significance level except for temperatures with four other variables in DJF (Table 2). Seasonal temperatures show the highest negative correlations with the SPEI (MAM, JJA and SON) and SPI (DJF). As expected, the seasonal PDSI series shows the highest positive correlations in all seasons with the Z-index and precipitation series with the SPI. The seasonal SPEI series indicates the highest correlations with SPI in all seasons except JJA; the same appears for the Z-index series with SPEI except DJF. The seasonal SPI series exhibits the highest correlations with JJA and SON precipitation, while in the two remaining seasons, it is the best correlated with the SPEI. The maxima of the highest correlation coefficients for all variables occur in JJA (0.991 between precipitation and SPI). The minima of the highest correlations appear in DJF (for temperature and precipitation), MAM (for SPI and PDSI) and SON (for Z-index). Temporal changes in the shared variability expressed with the running correlations show very similar features in all seasons as those for annual series (Fig. 2), and they are not shown here.

### 4.1.2 Series derived from phenological data
For Czech reconstructions based on phenological data (Možný et al., 2012, 2016a, 2016b), both temperature reconstructions agree in the warmest 30-year interval in 1991–2020 but differ in the coldest 30-year period: 1671–1700 in reconstruction for MAMJ from winter wheat harvest dates and 1835–1864 in reconstruction for AMJJA from grape harvest dates (Fig. 7). The first of this series also shows a statistically significant increasing linear trend (0.16°C/100 years). The AMJJA SPEI series exhibits the driest 30 years in 1991–2020, while the wettest period occurred at the beginning of the 20th century (1900–1929). For this series,

the driest period experienced much higher variability than the wettest, documented particularly by interquartile range. The means of all selected 30-year extreme periods differed significantly from the means of the entire 520-year period.

In Fig. 8, one can assess the agreement between the two temperature reconstructions derived from different documentary data (phenological series by Možný et al., 2012, 2016a versus temperature indices by Dobrovolný et al., 2010) on different time scales. Even if the overall correlations between the two types of reconstructions are quite high and significant (0.67 for series derived from winter wheat harvest dates and 0.82 for those derived from grape harvest dates), 31-year running correlations reveal that the common signal varies substantially over time. Generally, it is lower before 1800 CE when the two compared series are represented by reconstructed values. Very high and significant correlations were also found for a relatively long period from the second half of the 16th century to the mid-17th century, which could be perhaps related to the higher quantity and quality of the available documentary evidence.

Whereas the running correlations allow us to compare the common signal on the annual and decadal time scales, low-pass filtering of the series with the 60-year splines reveals common features of multidecadal variability (Fig. 8). The long-term trend is quite consistent for AMJJA temperatures derived from grape harvest dates and from temperature indices. In contrast, smoothed winter wheat harvest date series show much higher long-term variability compared to index-based reconstruction before 1800. The reconstruction from winter wheat harvest dates is well expressed, especially the period of low temperatures corresponding to the well-known Late Maunder Minimum of solar activity (1675–1715). This cold period is not as well expressed in the index-based temperature reconstruction because this central European reconstruction may partly smooth local effects.

## 4.2 Wavelet analysis

While strictly periodic components are typically not dominant in central European climate series beyond the annual time scales, the presence of noteworthy unstable periodicities has been previously reported for some climatic characteristics (e.g., Brázdil et al., 2012; Mikšovský et al., 2019). As seen from the wavelet spectra of individual Czech series (Fig. 9), there are indeed several period bands in which notable (and sometimes statistically significant) oscillations exist. In the case of multidecadal variability, periodicities of approximately 70–100 years appear in several signals, albeit rather intermittent in terms of amplitude. In the case of documentary-based data, these can be detected in both temperature and precipitation series, as well as in the series of drought indices (only SPEI shown here: wavelet spectra of SPI are generally similar to those of precipitation, while Z-index and PDSI resemble SPEI in their spectral structure). Presence of the *c.* 70-year periodicity is particularly pronounced during JJA in precipitation and drought index series, whereas its statistically significant manifestations in other seasons are limited to shorter subperiods. The existence of 70–100-year oscillations is also supported by their appearance in the wavelet spectra of temperature and SPEI series reconstructed from wheat and grape harvest dates (bottom row of Fig. 9), although, again, these test statistically significant only in a part of the 1501–2020 period.

On shorter time scales, periodic components in the Czech climate series are typically even more scattered. Most notably, in both indices-based and phenology-derived series, oscillations at periods of approximately 16–30 years are detected over some shorter subperiods. While these are typically statistically nonsignificant on their own over most of the analysis period, there are indications of interesting similarities to the spectral characteristics of several explanatory factors involved in our analysis. These are examined in more detail in Sect. 4.3.

## 4.3 Attribution analysis

A combination of regression analysis and wavelet transform was used here to identify and quantify links between reconstructions of Czech climatic characteristics and several potentially influential explanatory factors (for visualization of their temporal variability during 1501–2020 see Fig. 10; their mutual correlations are provided in Fig. S1 in the Supplement). The predictors used in our analysis (Sect. 2.3) exhibit only mild collinearity (with the strongest correlation detected between GHGRF and SOLAR, at $r = 0.45$). The results of linear regression (summarized in Fig. 11 through standardized regression coefficients and their confidence intervals) are therefore not substantively affected by variability shared by different predictors. Note that, unlike in prior analysis presented in Mikšovský et al. (2019), the El Niño – Southern Oscillation (ENSO) was not included among the explanatory factors due to largely negligible influence exhibited by the available ENSO reconstructions covering our target period. We also do not present results obtained separately for instrumental and pre-instrumental periods as was done in Mikšovský et al. (2019), because such division tends to magnify uncertainties pertaining to identification of slow-variable components in climatic time series. Furthermore, outcomes for PDSI are not shown due to the long memory component in this drought index, making proper pairing of predictand and predictors problematic without additional transformations.

As expected, due to the generally strong relationship between greenhouse gases forcing and temperatures worldwide, there is a prominent GHGRF-correlated component in the temperature series (corresponding to an approximately 1.8°C increase between 1501 and 2020). This link is also notable in the temperature-sensitive drought indices (SPEI, Z-index), most prominently during JJA and SON (Fig. 11a). In the precipitation data, the GHGRF-related trend is typically nonsignificant (except in MAM), and the direction of the respective link varies with season.

While there is a statistically significant association between solar irradiance and Czech climatic characteristics in a limited number of cases (particularly in SON for temperature – Fig. 11b), this relationship disappears when the slow-variability component is removed from the SOLAR series (i.e., when only solar variability at periods of approximately 11 years or shorter is used as a predictor). Considering also that cross-wavelet analysis suggests only an intermittent link between temperature and SOLAR and that mutual phases of the respective oscillations are highly variable in time (Fig. 12), the direct influence of solar activity in central Europe seems unlikely from our data, at least at decadal or shorter time scales.

The signature of volcanic activity is generally weak in the precipitation data (as well as in precipitation-dominated drought indices, especially SPI – Fig. 11c). There is, however, a clear (and statistically significant) tendency for colder conditions following major volcanic eruptions, manifesting through negative regression coefficients between temperature and volcanic aerosol optical depth. This link is strongest during JJA but nonsignificant during DJF and MAM.

Although the NAO represents one of the major weather drivers in central Europe, its effects are highly variable both seasonally and regarding the type of target variable. For temperature, a strong tendency towards warmer conditions is associated with a positive NAO phase in all seasons except JJA (Fig. 11d). For precipitation and drought indices, the links are typically weaker, with most significant responses detected for MAM and SON. The relationship between NAO and temperature is also detectable from the cross-wavelet spectra (Fig. 12) and wavelet coherence (Fig. S2 in the Supplement), with oscillations at periods of approximately 25 and 70 years being the most prominent and relatively consistent in terms of phase difference. Similar shared periodicities can also be found in relationships between NAO and precipitation or drought indices, albeit in slightly weaker form.

As shown in Mikšovský et al. (2019), there are notable links between variations in the central European climate and decadal and multidecadal oscillations in the northern Atlantic and northern Pacific. Expanding on these prior experiments, we used the detrended common component (AMO+PDO) and difference (AMO-PDO) of temperatures in the AMO and PDO regions provided by Mann et al. (2009) as potential explanatory variables here. In the case of shared AMO and PDO variability, linear regression reveals a significant link to Czech temperature during all seasons except DJF (Fig. 11e). On the other hand, precipitation and all drought indices exhibit a relationship to differences in AMO and PDO phases, most pronounced during the SON season (Fig. 11f). The cross-wavelet analysis further suggests the stability of the temperature to the AMO+PDO link around the period of approximately 70 years, at least from approximately 1650 CE on (Figs. 12 and S2). Another region of spectral similarity appears around a period of 25 years, but the relationship is more intermittent and unstable in terms of phase shift. For the link between precipitation and AMO-PDO signals, the primary band of shared periodicities seems to be located between *c*. 8 and 16 years, but again, some variations in phase shifts do appear.

**4.4 Spatiotemporal representativeness of Czech reconstructions**

To show the spatiotemporal representativeness of Czech reconstructions of selected climate variables, they were compared with related gridded reconstructions for Europe. The seasonal central European temperature series by Dobrovolný et al. (2010), compared with European temperature reconstructions by Luterbacher et al. (2004) in the 1501–2002 period, shows the highest correlation coefficients (>0.60) in the large area extending from the British Isles to eastern central Europe in the west-east direction and from south Scandinavia to the Mediterranean in the north-south direction (Fig. 13). This area is the largest during DJF, when it extends far to eastern Europe, and the smallest in SON, when it does not cover a part of eastern central Europe (particularly Poland). In JJA, the highest correlations also extend over the whole British Isles, northwestern part of the Iberian Peninsula, Apennine Peninsula and south Scandinavia. Comparing temporal consistency between the two types of series (series for central Europe from Luterbacher et al., 2004 and Xoplaki et al., 2005, was calculated for the window limited by geographic coordinates 45°N–54°N and 5°E–23°E), it shows very high 31-year running correlation coefficients during the entire 500 years except a steep drop in correlations close to the significance level in JJA temperatures approximately around 1750 CE (Fig. 14a). The overall statistically significant correlation coefficients for the entire analysed period are the highest for DJF (0.94), while in the remaining seasons, they are 0.88 (MAM, JJA) and 0.89 (SON).

Compared to temperatures, the comparison of seasonal Czech precipitation reconstructions by Dobrovolný et al. (2015) with gridded European precipitation reconstructions by Pauling et al. (2006) for the 1501–2000 period suffers from great spatial variability of precipitation totals (Fig. 15). Although a broad belt of positive correlations extends from western to eastern Europe, the areas with highest correlations are much smaller, oriented rather to the area located westerly of the Czech territory. The Czech precipitation reconstruction is most representative in SON, while the weakest agreement appears in MAM. The 31-year running correlations between the two types of series generally decrease from the beginning of the 16th century to the mid-first half of the 18th century (even with values below the significance level for MAM and SON), with an increasing trend afterwards (Fig. 14b). However, in addition to these trends, some remarkable drops or increases in correlation coefficients also appear (such as a drop in the beginning of the 20th century in MAM totals or an increase approximately around 1725 CE in JJA totals). The overall correlation coefficient is the highest in JJA (0.67) and the smallest in MAM (0.50), but statistically significant in all seasons.

Due to the lack of existing gridded European reconstructions of drought indices from documentary data, the JJA scPDSI series (Brázdil et al., 2016) was compared with the same European series but reconstructed from tree rings in OWDA (Cook et al., 2015) during the 1501–2012 period. As follows from Fig. 16, there is only weak spatial consistency with larger positive correlations around the Czech territory, extending to southeast and westerly as far as France, and exhibiting rather a spotty character. It is also reflected in 31-year running correlations with the series of central European windows from Cook et al. (2015), where the drop in correlations appears in the second half of the 16th century and particularly during the 18th century, with values deeply under the 0.05 significance level (Fig. 14c). This is reflected in the low overall correlation coefficient between the two series, achieving only 0.40 (but statistically significant).

Because the Czech climate reconstructions are spliced from "reconstructed" and "instrumental" parts (see Sect. 2.1 for details), questions about the effects of these two parts on spatial representativeness may appear. For this reason, temperature reconstruction was compared spatially separately for two 150-year-long periods from both mentioned parts of the series (Fig. 17). Correlations are high and significant for both parts of the series covering a large area of Europe with latitudes from *c*. 60°N to the south and longitudes from *c*. 25°E to the west. Moreover, the area of significant spatial correlations was quite similar for all seasons (not shown). On the other hand, it is necessary to say that a very preliminary version of the Czech temperature/precipitation index series compiled from a significantly lower density of documentary evidence at that time was used in corresponding gridded European reconstructions by Luterbacher et al. (2004), Xoplaki et al. (2005) and Pauling et al. (2006).

## 5 Discussion

### 5.1 Climate fluctuations and European context

Proxy-based reconstructions reflect the main features of climate fluctuations. However, they can also be affected by the quality and quantity of proxies. In addition, methods of chronology compilation and data analysis may play a role. While in the case of natural proxies (e.g. tree rings) these non-climatic factors may be controlled to some extent during the process of standardization, in the case of documentary evidence it is more problematic for obvious reasons (see e.g. detail discussion in Brázdil et al., 2010).

With respect to these facts, mutual comparison of different climate reconstructions is an important tool to highlight strengths and weaknesses of individual reconstructions and outline possible reasons for some peculiarities in their variability. In this study, the comparison was based on the correlation analysis as well as on the direct comparison of smoothed series to highlight common variability on decadal and multidecadal scales (see Figs. 2, 8, and 14). The following text summarizes the main features of such comparison that have been explained in detail in the original "reconstruction" papers. Moreover, we are trying to explain possible reasons that may be responsible for the loss of common signals in some periods.

As for temperatures reconstructed from documentary indices, very high and statistically significant correlations follow from the comparison of central European temperature series by Dobrovolný et al. (2010) with gridded multiproxy European reconstructions of seasonal temperatures by Luterbacher et al. (2004) and Xoplaki et al. (2005), recalculated only for central European window (Fig. 14a). But around the mid-18th century there appeared a deep decline in correlations for JJA temperatures, discussed already by Dobrovolný et al. (2010). One of its reason could be the quality and quantity of available data. The reconstruction has been based on documentary-derived series of temperature indices for Germany, Switzerland and the Czech Lands. However complete series of German indices have been available only prior to1760 and Swiss indices prior to the 1810s, while the Czech

indices continued to the mid-19th century. This could result in lower temperature variability (see Fig. 14 in Dobrovolný et al., 2010) and subsequently in a lower coherence with other proxy-based reconstructions in this period.

However, a closer look at relationships between the two compared reconstructions in Figure 14a reveals another problem. Calculation of JJA temperature differences between reconstructions by Dobrovolný et al. (2010) and Luterbacher et al. (2004) shows positive differences before the mid-18th century and negative afterward. This shift is responsible for a sharp decrease in running correlations. In order to evaluate this inconsistency, differences of these two series with regard to completely independent JJA multiproxy temperature reconstruction for the Alps by Trachsel et al. (2012) were calculated. For better comparison, the series were first transformed to have a mean of zero and a standard deviation of one. While the differences with the series by Dobrovolný et al. (2010) were distributed more or less randomly around zero, the differences with the Luterbacher et al. (2004) series showed the same patterns as described above: positive differences before the 1750s (i.e., higher temperatures by Trachsel et al., 2012) and negative differences afterward. This indicates that the problem of lost coherence around the 1750s in Fig. 14a cannot be attributed to Dobrovolný et al. (2010) reconstruction.

As for series derived from phenological data, MAMJ temperatures reconstructed from winter wheat harvest dates were compared with 11 late spring and summer temperature series in central Europe (see Fig. 6 in Možný et al., 2012). Better coherence was found with documentary-based and biophysically-based reconstructions (harvest dates) than those based on tree-rings. A significant drop in correlations appeared particularly in the second half of the 17th century and around the 1750s. This may be partly related to the problem in the data quality of the winter wheat harvest dates. These dates had to be recalculated from the harvest dates of other available cereals in periods when the winter wheat dates were not available. The distinct role may be attributed to the "social bias" in data related to the complicated social and political situation in the country (see discussion related to those periods in Možný et al., 2012, and also Fig. 8a in the current study).

Similarly, AMJJ temperatures reconstructed from grape harvest dates were compared with 17 European temperature reconstructions based on temperature indices derived from documentary data, grape harvest dates, tree-rings, and multiproxies (see Fig. 9 in Možný et al., 2016a). Possible inconsistencies were found in the first half of the 16th century, around 1650, 1750, and 1900. Four periods with potential "social bias" were identified in the last decades of the 16th century and then in the 1640s–1670s, 1750s–1780s, and 1850s–1910s.

The comparison seems to be more problematic in the case of precipitation, characterised by high spatiotemporal variability. For example, less spatially homogeneous Czech JJA precipitation totals were plotted against six similar European precipitation reconstructions (see Fig. 9 in Dobrovolný et al., 2015). Periods of quite similar precipitation fluctuations were revealed particularly in the first half of the 16th century, in the 1630s and 1710s (dry decades), and approximately in the 1590s, 1690s, 1730s and 1810s (wet decades).

Documentary-based reconstructions of drought indices in the Czech Lands were correlated against six different European drought series (see Fig. 6 in Brázdil et al., 2016). The overall patterns were the same as in Figure 14c in this study. While there was a good agreement especially in the first half of the 16th and the 17th centuries, a drop in common variance appeared in the second half of the 16th century, in the 1650s–1750s and after the 1950s.

Differences between reconstructions and loss of coherence between them may also result from a natural climate variability. This applies especially for those covering a slightly different spatial domain or those reconstructing climate variables characterized by high spatial variability. As discussed in more detail in Možný et al. (2016a), some periods (e.g., Maunder

minimum in 1675–1715 – Frenzel et al., 1994) can be characterized with a higher frequency of meteorological extremes of the regional extent. Their more frequent occurrence in some regions may be conditioned dynamically (i.e. by different circulation patterns – see e.g. Wanner et al., 1995) and thus may be responsible for higher spatial climate variability and subsequently for lower correlations in comparison to related series on a central European scale.

An interesting aspect of lost common signal manifested by a decrease in running correlations below the 0.05 significance level can also appear in the "instrumental part" of the reconstructed series as documented in Fig. 2a. Running correlations of annual temperatures with other five climate variables are highly significant from the 16th century up to the early 19th century. These negative correlations are physically consistent as they show that higher temperatures usually correspond to low precipitation and *vice versa*. Approximately from the mid-19th to the mid-20th centuries correlations among all compared series are not significant. Despite the fact, that annual means express some mixture of different seasonal patterns, this gradual loss of common signal may be interpreted as follows. The fact, that before the 19th century the series are reconstructed from dependent (and thus less variable) temperature and precipitation indices, can be reflected in significant correlations. The instrumental parts of series (target data) are mutually less dependent and more variable than indices. The same patterns as in annual values (Fig. 2a) are well expressed also in SON series and partly in MAM and JJA series, while they do not occur in DJF series (non-significant correlations over the whole period) (not shown). The stronger common signal (significant negative correlation) occurring during the last decades can be attributed to a clearly expressed opposite tendency of rising temperatures and decreasing drought indices. The same pattern does not change even when correlating the detrended series or when changing the length of the window, for which running correlations were calculated.

## 5.2 Climate variability and forcings
While climate reconstructions based on documentary data exhibit distinct interannual and interdecadal variability, some doubts appear regarding the expression of low-frequency (long-term) signals in such series (e.g., Brázdil et al., 2010). In our current analysis, a possible indication of different representations of long-term variability comes from the results of the wavelet transform. Although spectra of univariate documentary-based Czech series do not exhibit a clear systematic tendency towards higher amplitudes of multidecadal oscillations in any specific subperiod (Fig. 9), diminished powers of shared oscillations in cross-wavelet spectra do appear for some of the explanatory variables, particularly around the 70–100-year period band (Fig. 12). On the other hand, such behaviour may be related to specific features of the explanatory variables themselves, particularly lower variances displayed by the NAO and AMO+PDO series in the early parts of the 1501–2020 period (Fig. 10). The phenological data provide a somewhat different representation of long-term oscillations in the temperature and drought-index series, with notable contrast between the early and later parts of the analysis period and peculiar differences between cereal- and grape-based reconstructions (Fig. 9). While this heterogeneity may be partly climatic in origin or related to crop-specific responses to particular weather patterns, variations in the geographical structure of growing locations or changes in cultivars grown (Možný et al., 2012, 2016a, 2016b) likely play a considerable role as well. Even so, the presence of distinct spectral similarities between indices- and phenology-based reconstructions supports the existence of *c.* 70–100-year oscillations affecting the Czech climate, despite discrepancies in their exact timing and amplitude.

The problem of potential misrepresentation of low-frequency variations particularly concerns the expression of temperature/precipitation patterns in the form of different ordinary

degree scales used for the creation of a series of temperature/precipitation indices that are less sensitive to characterizing particularly extreme values. It is well expressed in long-term trends of the analysed 520-year series, where no statistically significant trends appear in seasonal and annual precipitation (*cf*. Brázdil et al., 2021 from 1961 CE) and drought indices series,

only in temperature reconstructions, where it is mainly the effect of sudden temperature increase from the 1970s (*cf*. Zahradníček et al., 2021 from 1961 CE). It appears not only in the reconstruction based on temperature indices (Dobrovolný et al., 2010) but also in those derived from phenological data (Možný et al., 2012, 2016a). On the other hand, in reconstructions based on phenological data, non-homogeneities due to "social bias" may

appear. For example, Možný et al. (2012) considered this aspect in connection with the importantly warmer first half of the 16th century (for example, the use of the sickle for cutting requested more time, i.e., harvests started earlier) and importantly cooler the second half of the 17th century (total devastation of the Czech Lands after Thirty-Year War, coinciding with the cold Maunder Minimum period – see, e.g., Frenzel et al., 1994) in MAMJ reconstruction

from winter wheat harvest dates. The earliest start of harvests in 1517–1542 CE, even comparable to 1971–2010, was confirmed by Brázdil et al. (2019), analysing long-term changes in the agricultural cycle in the Czech Lands.

The presence of linear trends, detected especially for the temperature series and its drought-related derivates, can be approximated very well by the variations in greenhouse

gases concentrations and the resulting changes in radiative forcing. Despite this good formal match, note that statistical methods alone are unable to reliably confirm the causal nature of this relationship between long-term trends, and other approaches (such as simulations by dynamical models) are needed to verify causality.

Similar caution is needed in the case of solar activity: while the regression analysis

suggested the possibility of a relationship to Czech temperatures, this link vanished after the slow-variable component was removed from the SOLAR series (which eliminated aliasing between GHGRF and SOLAR signals). It should be emphasized, however, that our (strictly linear) analysis does not exclude the possibility of more complex interactions between central European climate and solar activity, possibly detectable by more general methods.

Unlike changes in solar activity, volcanic activity leaves a distinct imprint in the reconstructed temperature series. Cooling following major volcanic material ejections into the stratosphere is most notable during JJA; on the other hand, it is only borderline statistically significant in the temperature-sensitive drought indices (especially SPEI) and not detectable from the precipitation series.

Unsurprisingly, a strong effect of NAO was detected in most of the Czech series analysed, but the strength of its impact varied seasonally (with JJA exhibiting the weakest connection). Prominent components of this relationship seem to be tied to periodicities of approximately 70 years and 25 years, although the respective links are not completely stable in time.

Our analysis, involving temperature variability in the AMO and PDO regions as explanatory factors, has confirmed the distinct influence of both shared AMO and PDO variability (identified especially in the Czech temperature series) and their difference (significantly influencing Czech precipitation and drought indices). The results of cross-wavelet analysis suggest that this AMO/PDO impact may be related to shared periodic

oscillations in the *c*. 70- to 100-year period band. Other spectral similarities (although manifested in a less coherent fashion) have also been detected over the approximately 16- to 32-year period band (especially for the common AMO+PDO variability) and 8- to 16-year band (for the AMO-PDO difference). However, substantial variance in mutual phases revealed by the wavelet spectra suggests that the nature of these potential links goes beyond

simple linear responses, and a more complex analytical approach may be needed to fully unravel them.

Finally, note that the outcomes of the attribution analysis may also be subject to specific properties of the explanatory variables used, particularly in case of the reconstructed indices of internal climate variability modes (NAO, AMO, PDO). This issue has been previously investigated in Mikšovský et al. (2019), where multiple independent reconstructions were used for each of these indices (including NAO reconstructions by Trouet et al., 2009 or Ortega et al., 2015, AMO reconstruction by Gray et al., 2004, and PDO reconstructions by MacDonald and Case, 2005 or Shen et al., 2006). The series based on data by Luterbacher et al. (2001) and Mann et al. (2009) were shown to carry the relatively strongest link to central European climate variability, and were therefore employed in this current analysis. Even so, the problem of predictor-related robustness of the attribution analysis remains an essential one and an issue worthy of revisiting, especially when new relevant proxy-based data arrive in the future.

**6 Conclusions**

From the analysis of 520-year series of reconstructed temperature, precipitation and drought indices based on documentary data and instrumental observations in the Czech Lands, the following conclusions can be summarized:

(i) All Czech temperature reconstructions regardless of the season and the proxy data used show the exceptionality of high temperatures in the last three decades in the context of the past 500 years. On the other hand, the coldest 30-year periods occurred before the 1850s in all seasons.

(ii) Temperature reconstructions compiled from the phenological proxies better capture the long-term trends compared to temperature index-based reconstruction. However, they also show some shorter periods of lower temperature variability, which may be related to nonclimatic (anthropogenic) factors.

(iii) 520-year temperature and drought indices confirm extremeness of 1991–2020 as the warmest and driest 30-year period. While only annual and seasonal temperature series experience statistically significant long-term linear trends, a better match of long-term temperature components was found through regression against greenhouse gases radiative forcing. An increase in temperature is the key factor of increasing dryness in recent decades, while precipitation totals remain relatively stable with evident year-to-year and decadal variability.

(iv) While seasonal central European temperature reconstruction shows high spatiotemporal representativeness for the broad belt of Europe extending from western to eastern Europe and from the Mediterranean to south Scandinavia (with some territorial differences among seasons), seasonal precipitation reconstructions importantly decrease as a feature of high spatiotemporal variability in precipitation.

(v) Our analysis confirmed the influence of volcanic activity (manifested in the temperature series, especially in JJA) and the NAO index (exhibiting a strong influence in all seasons except JJA) on multicentury variability in the central European climate. Furthermore, components correlated with AMO- and PDO-related multidecadal oscillations were detected in both the temperature and precipitation series. While the temperature variations are tied mostly to the shared common component of the AMO and PDO (and thus general temperature variations across the Northern Hemisphere), precipitation (as well as all drought indices in our analysis) seems to be primarily affected by the difference between temperatures in the AMO and PDO regions. Similarities between AMO/PDO oscillations and multidecadal variability in central Europe are particularly noticeable in the *c.* 70–100-year period bands, although the relationship is not stable throughout the entire 1501–2020 period.

(vi) While various prominent linear structures and relationships were detected for our target series, complexity of some of the links suggests potential for additional improvement from application of more specialized methods, better suited to deal with non-stationarities, non-linearities and uncertainties in the data. Future development and application of such techniques may reveal additional influences, contributing to recorded climate variability in central Europe.

**Data availability.** The temperature series of central Europe are available at https://www.ncei.noaa.gov/access/paleo-search/study/9970. Precipitation series of the Czech Republic, the Czech temperature and precipitation reconstructions based on phenological data and drought indices series are available from the corresponding authors or the relevant publications. Other datasets were obtained from following databases: http://drought.memphis.edu/OWDA/Default.aspx for scPDSI; https://www.ncdc.noaa.gov/paleo-search/study/ 6342 for precipitation; and https://www.ncdc.noaa.gov/paleo-search/ study/6288 for air temperature. Series of explanatory variables were obtained from public climate databases (such as ClimExp – https://climexp.knmi.nl/) or from supplements of respective papers referenced in the text.

**Author contributions.** RB designed and together with JM and PD wrote the paper with contributions from all coauthors. PD analysed fluctuations of the Czech series and their representativeness in the European context. JM performed attribution analysis and PP wavelet and wavelet coherency analysis. MM, MT and JB contributed to the reconstructed temperature and drought index series. All authors have read and commented on the latest version of the paper.

**Competing interests.** The authors declare that they have no conflict of interest.

**Special issue statement.** This article is part of the special issue "International methods and comparisons in climate reconstruction and impacts from archives of societies". It is not associated with a conference.

**Acknowledgements.** RB, PD, MT and JB were supported by the Ministry of Education, Youth and Sports of the Czech Republic for the SustES – Adaptation strategies for sustainable ecosystem services and food security under adverse environmental conditions (project no. CZ.02.1.01/0.0/0.0/16_019/0000797) and JM by the Czech Science Foundation (grant no. 19-16066S). R library *biwavelet* was used for calculation and visualization of wavelet spectra. AJE is acknowledged for English style corrections.

**Financial support.** This research has been supported by the Ministry of Education, Youth and Sports of the Czech Republic (grant no. CZ.02.1.01/0.0/0.0/16_019/0000797) and by the Czech Science Foundation (grant no. 19-16066S).

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

**Table 1.** The warmest and driest (a) and the coldest and wettest (b) 30-year periods in annual and seasonal series of climate variables (CV) in the Czech Lands in 1501–2020 CE: T – temperature, P – precipitation, SPI, SPEI, Z-in (Z-index) and PDSI – drought indices.

(a) Warmest (T) and driest (P, SPI, SPEI, Z-in, PDSI)

| CV | Annual | DJF | MAM | JJA | SON |
|---|---|---|---|---|---|
| T | 1991–2020 | 1988–2017 | 1991–2020 | 1991–2020 | 1991–2020 |
| P | 1699–1728 | 1725–1754 | 1773–1802 | 1700–1729 | 1605–1634 |
| SPI | 1704–1733 | 1680–1709 | 1773–1802 | 1700–1729 | 1605–1634 |
| SPEI | 1990–2019 | 1680–1709 | 1989–2018 | 1990–2019 | 1605–1634 |
| Z-in | 1990–2019 | 1991–2020 | 1991–2020 | 1990–2019 | 1990–2019 |
| PDSI | 1991–2020 | 1991–2020 | 1991–2020 | 1991–2020 | 1991–2020 |

(b) Coldest (T) and wettest (P, SPI, SPEI, Z-in, PDSI)

| CV | Annual | DJF | MAM | JJA | SON |
|---|---|---|---|---|---|
| T | 1829–1858 | 1572–1601 | 1832–1861 | 1569–1598 | 1757–1786 |
| P | 1912–1941 | 1555–1584 | 1885–1914 | 1568–1597 | 1910–1939 |
| SPI | 1912–1941 | 1555–1584 | 1894–1923 | 1568–1597 | 1910–1939 |
| SPEI | 1569–1598 | 1555–1584 | 1873–1902 | 1569–1598 | 1910–1939 |
| Z-in | 1912–1941 | 1898–1927 | 1876–1905 | 1569–1598 | 1887–1916 |
| PDSI | 1913–1942 | 1913–1942 | 1888–1917 | 1913–1942 | 1912–1941 |

**Table 2.** Pearson correlation coefficients between seasonal series of temperature (T), precipitation (P) and drought indices (SPI, SPEI, Z-index, PDSI) in the Czech Lands during the 1501–2020 period (coefficients expressed in italics in brackets are statistically nonsignificant at the 0.05 significance level; all other coefficients are statistically significant).

DJF

| Variable | T | P | SPI | SPEI | Z-index | PDSI |
|---|---|---|---|---|---|---|
| T | x | (*0.067*) | 0.365 | (*0.125*) | (*0.081*) | (*-0.063*) |
| P | -0.309 | x | 0.831 | 0.831 | 0.598 | 0.278 |
| SPI | -0.348 | 0.894 | x | 0.956 | 0.662 | 0.268 |
| SPEI | -0.703 | 0.814 | 0.899 | X | 0.703 | 0.325 |
| Z-index | -0.675 | 0.790 | 0.887 | 0.971 | x | 0.795 |
| PDSI | -0.430 | 0.407 | 0.431 | 0.525 | 0.615 | X |

MAM

JJA

| Variable | T | P | SPI | SPEI | Z-index | PDSI |
|---|---|---|---|---|---|---|
| T | x | -0.561 | -0.563 | -0.778 | -0.760 | -0.558 |
| P | -0.235 | x | 0.991 | 0.943 | 0.932 | 0.583 |
| SPI | -0.241 | 0.985 | x | 0.950 | 0.939 | 0.587 |
| SPEI | -0.552 | 0.925 | 0.937 | X | 0.974 | 0.650 |
| Z-index | -0.545 | 0.848 | 0.851 | 0.922 | x | 0.717 |
| PDSI | -0.387 | 0.410 | 0.406 | 0.488 | 0.721 | X |

SON

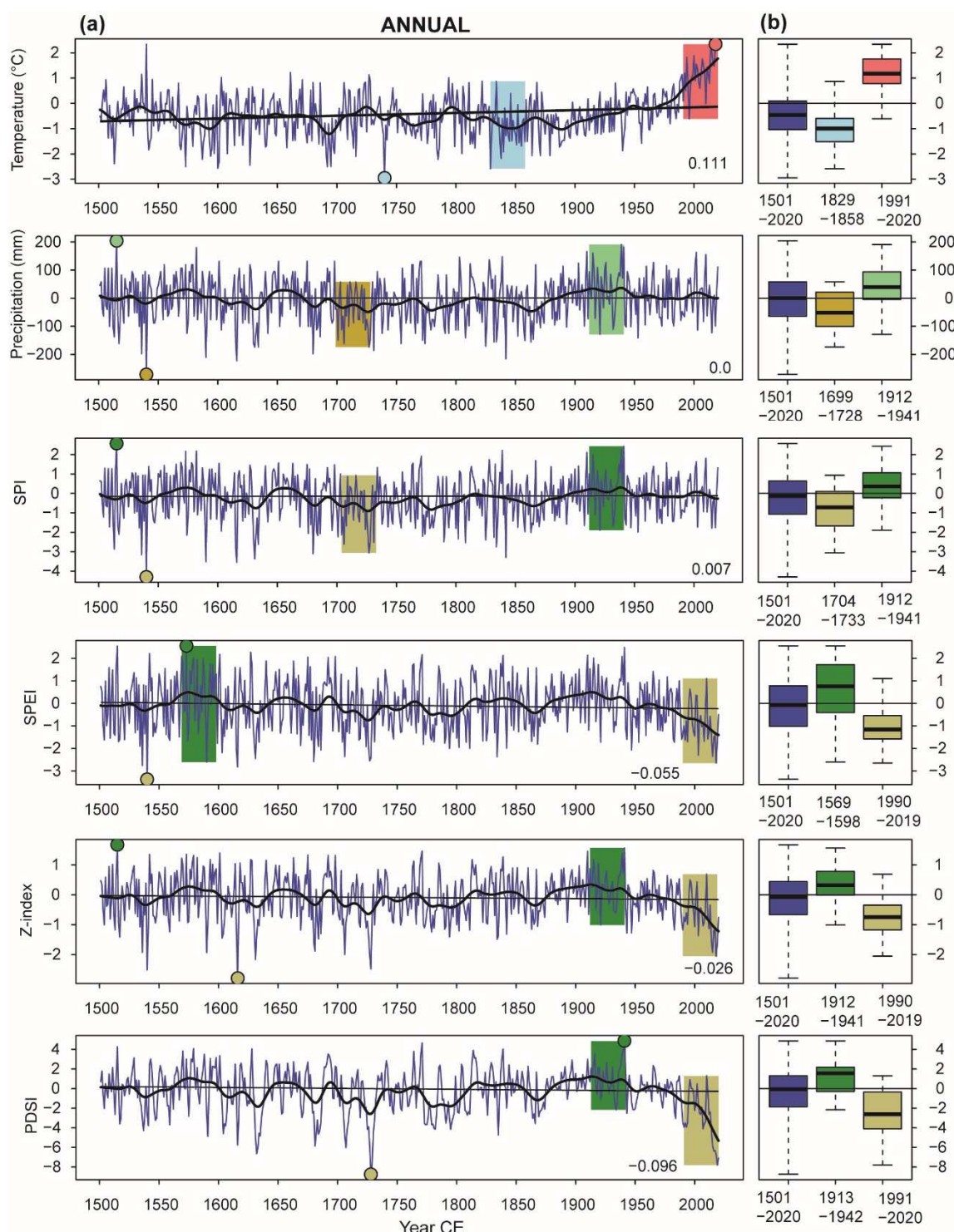

**Figure 1.** Selected annual climate variables in the Czech Lands during the period of 1501–2020 CE: (a) fluctuations smoothed by 30-year Gaussian filter with linear trends and their numeric values (temperature: °C/100 years, precipitation: mm/100 years, drought indices: index value/100 years), and extreme 30-year periods of each series indicated by coloured
10    bands and the lowest and highest values of series by small circles; (b) box plots (median, lower and upper quartile, minimum and maximum) for 1501–2020 and two most extreme 30-

year periods. The temperature and precipitation series are expressed as deviations with respect to 1961–1990.

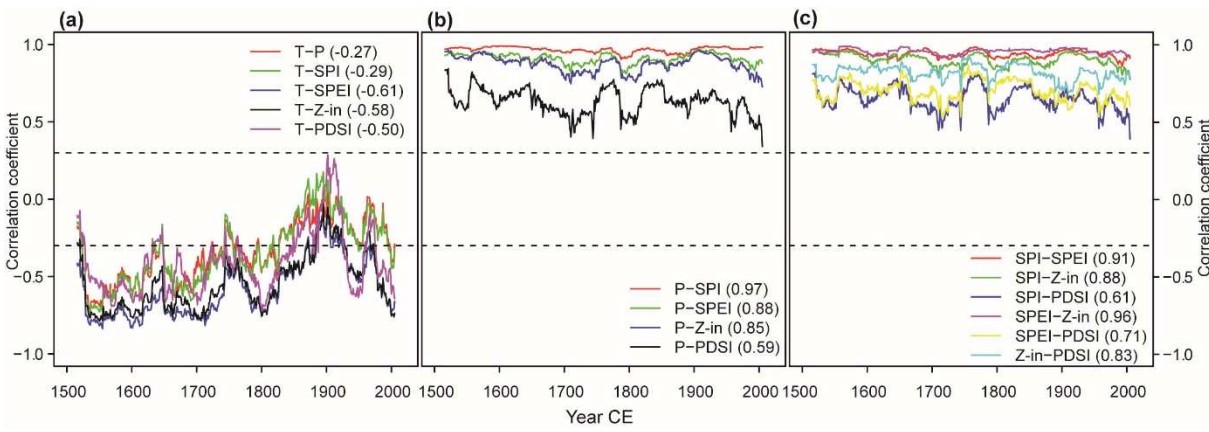

**Figure 2.** 31-year running correlation coefficients between annual series of (a) temperature (T), (b) precipitation (P) and (c) drought indices (SPI, SPEI, Z-index, PDSI) in the Czech Lands in the 1501–2020 period. Correlation coefficients for the whole period are in brackets. Dashed lines indicate 0.05 significance levels: correlations above/below these levels are statistically significant.

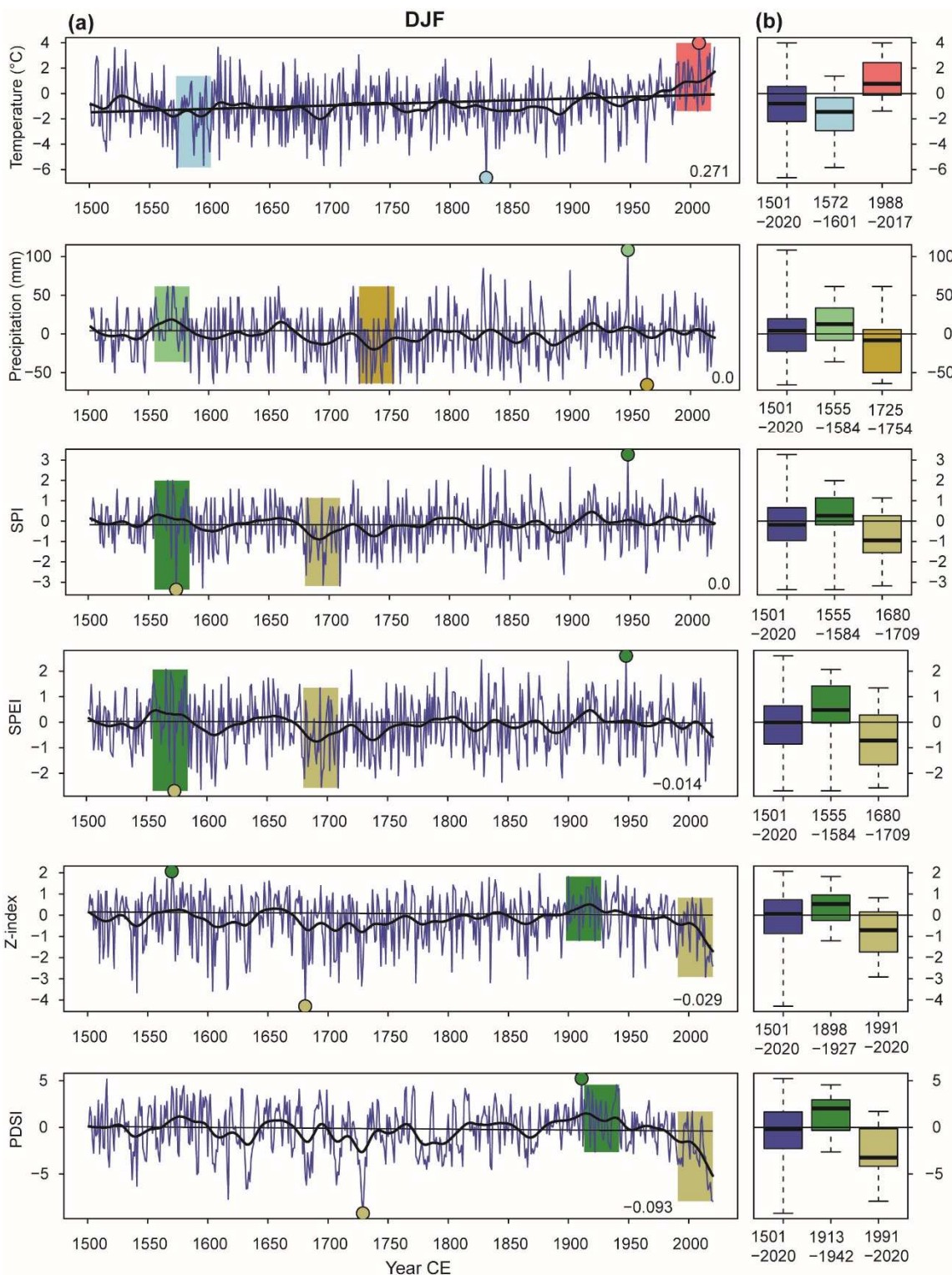

**Figure 3.** See text in Figure 1, DJF.

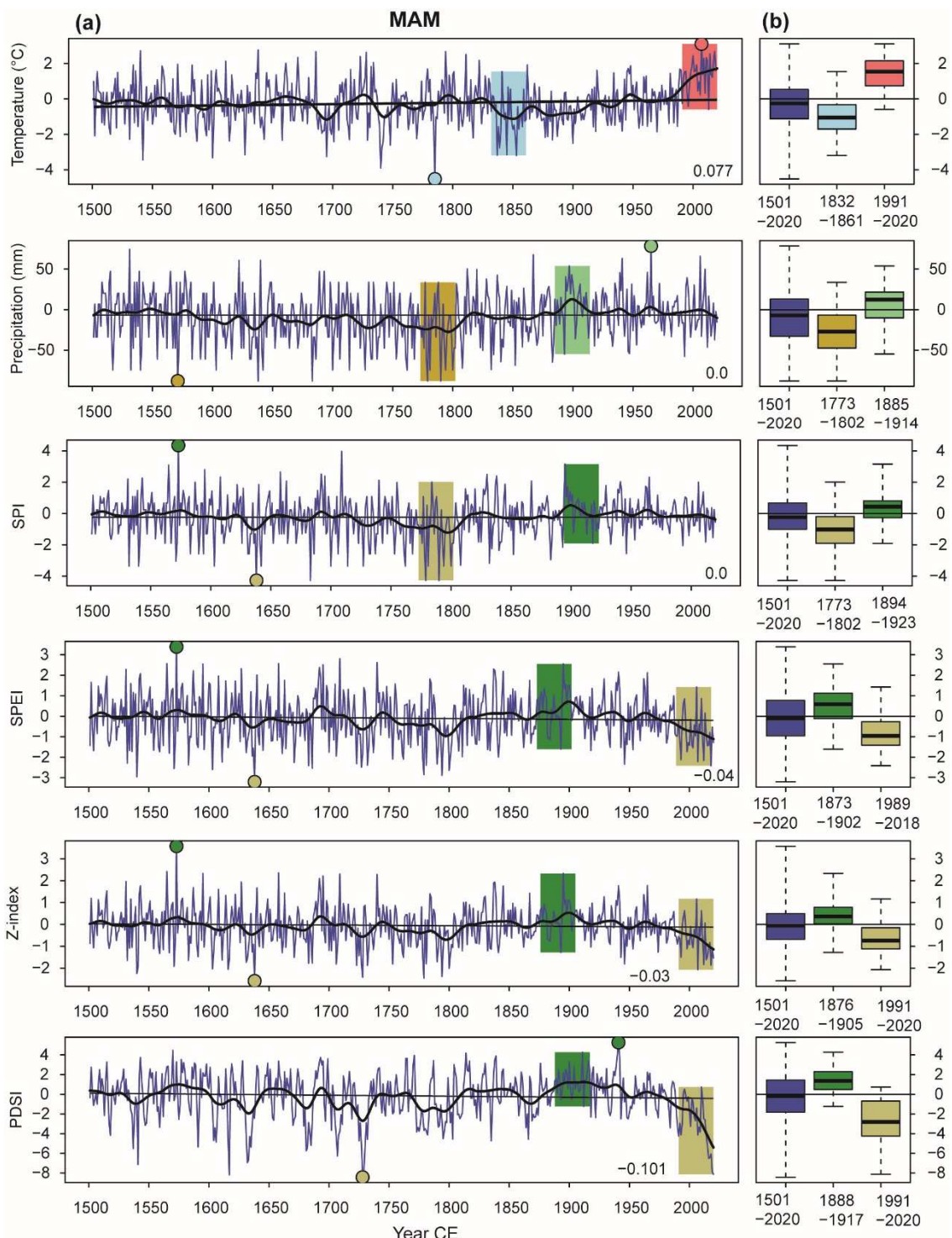

**Figure 4.** See text in Figure 1, MAM.

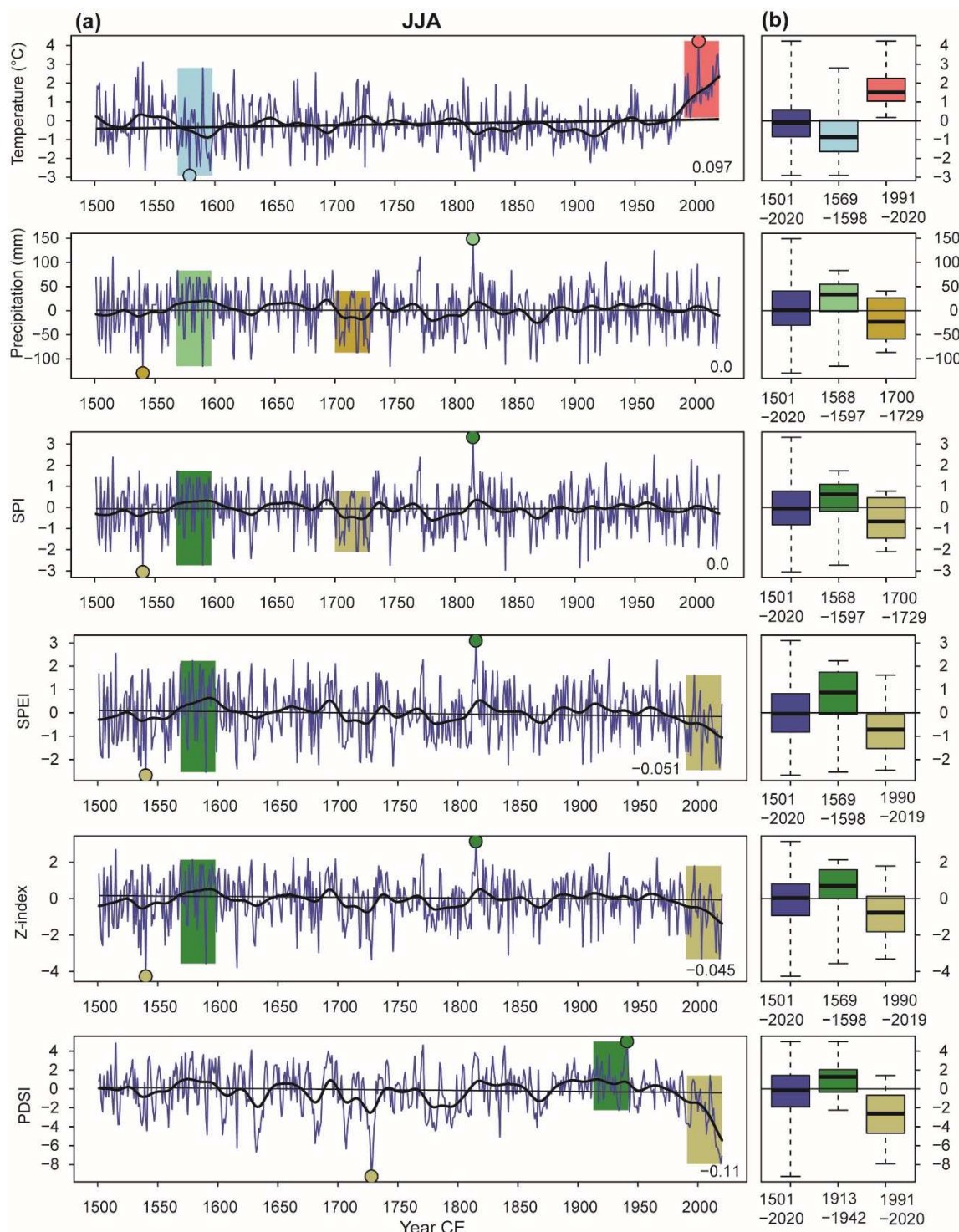

**Figure 5.** See text in Figure 1, JJA.

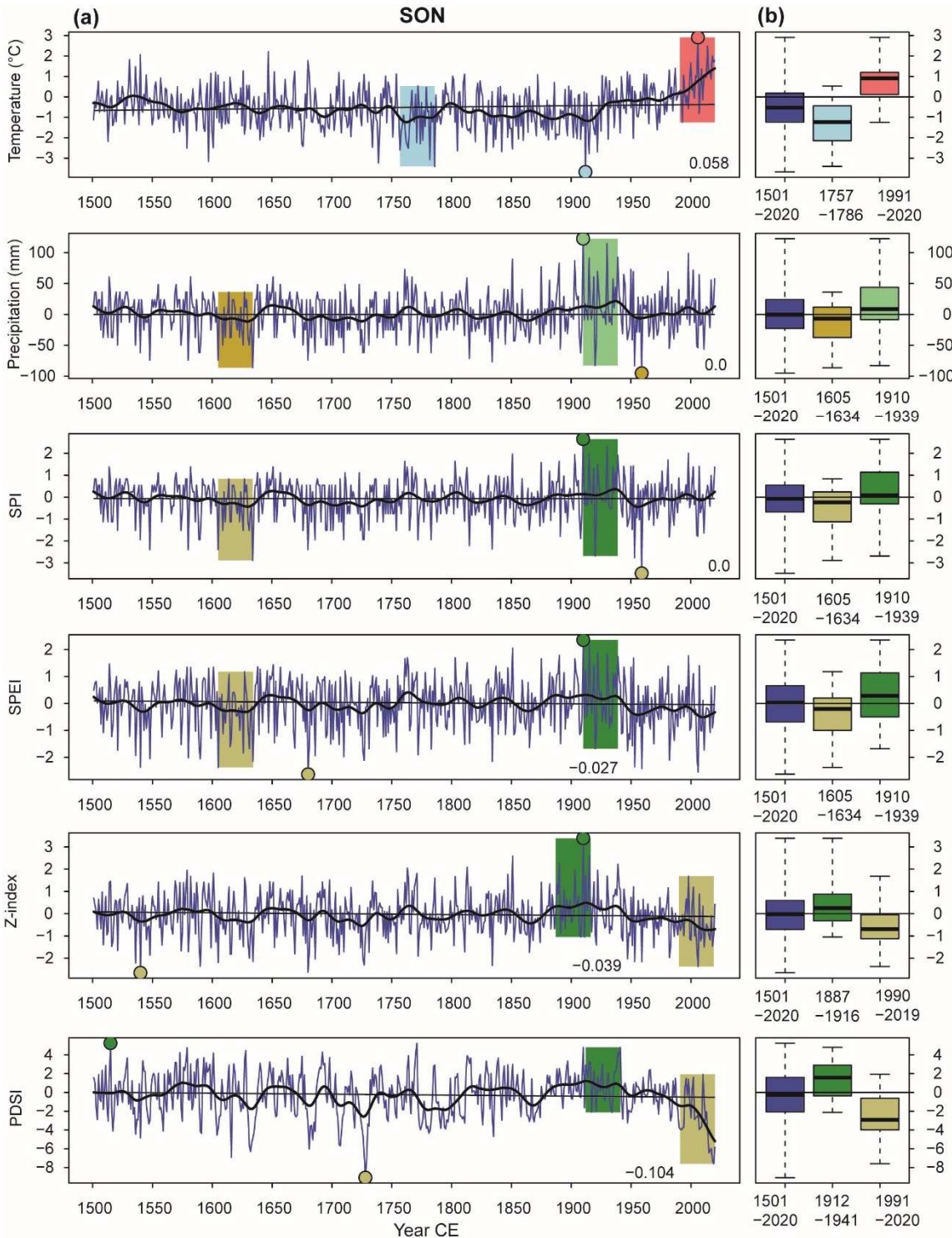

**Figure 6.** See text in Figure 1, SON.

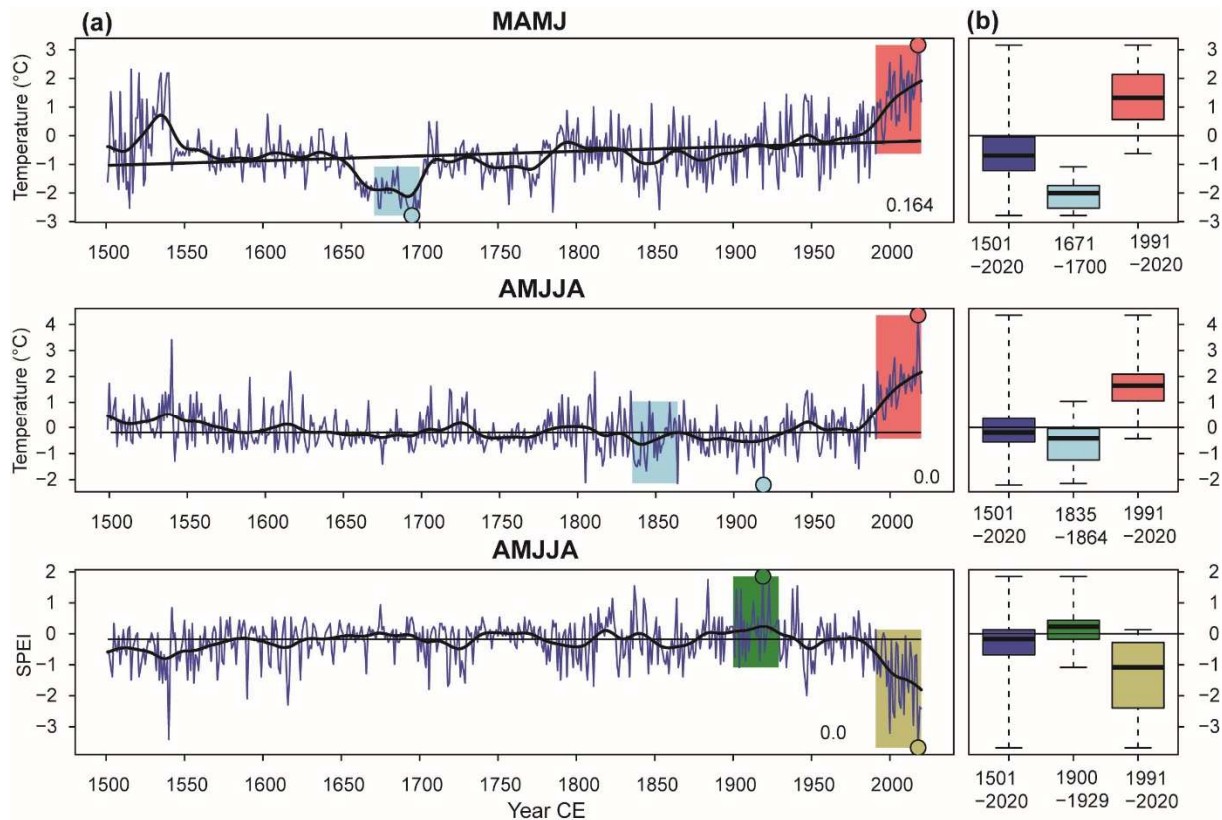

**Figure 7.** (a) Variability of MAMJ mean temperatures reconstructed from the winter wheat harvest dates (Možný et al., 2012), AMJJA mean temperatures and SPEI reconstructed from the grape harvest dates (Možný et al., 2016a, 2016b) in the Czech Lands during the period of 1501–2020 CE. Annual values are completed with the 30-year Gaussian filter, linear trends and their numeric values (temperature: °C/100 years, SPEI: index value/100 years); extreme 30-year periods of each series are indicated by coloured bands and the lowest and highest values of series by small circles. (b) Box plots express the median, lower and upper quartile, minimum and maximum for 1501–2020 and the two most extreme 30-year periods. Temperature series are expressed as deviations with respect to the 1961–1990 period.

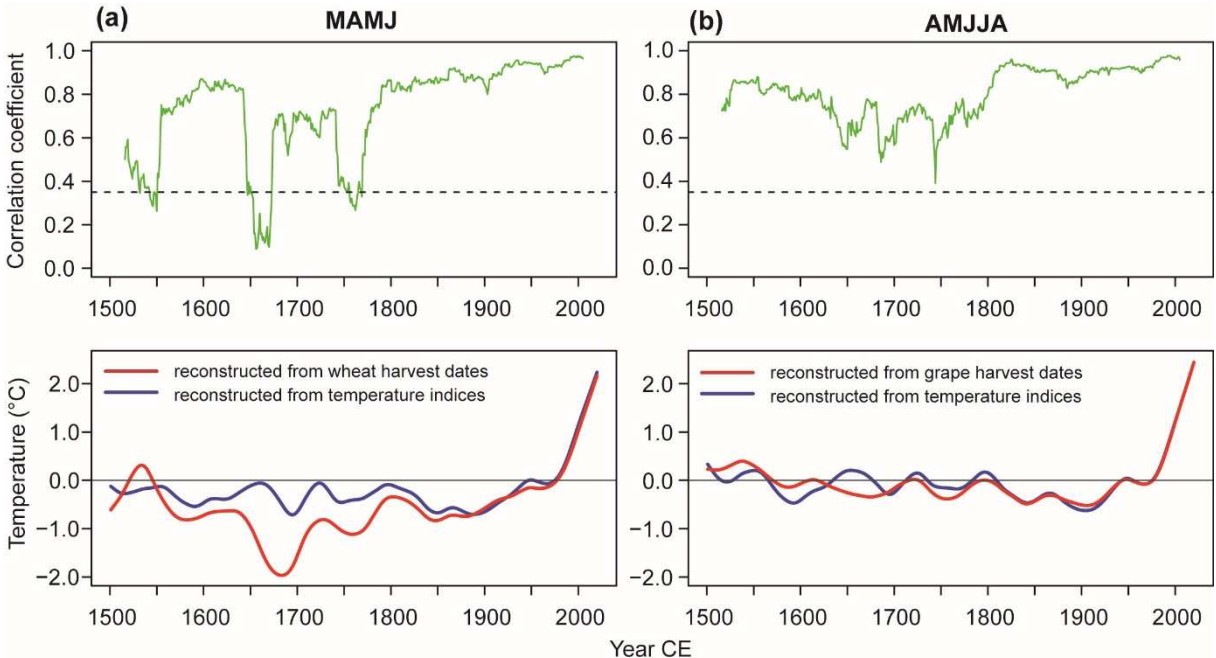

**Figure 8.** 31-year running correlations (top) and low-frequency signal expressed as smoothed series by the 60-year spline function (bottom) compared to MAMJ temperatures reconstructed from the wheat harvest dates (Možný et al., 2012) and those reconstructed from temperature indices (a); (b) the same as (a) but for AMJJA temperatures reconstructed from the grape harvest dates (Možný et al., 2016a).

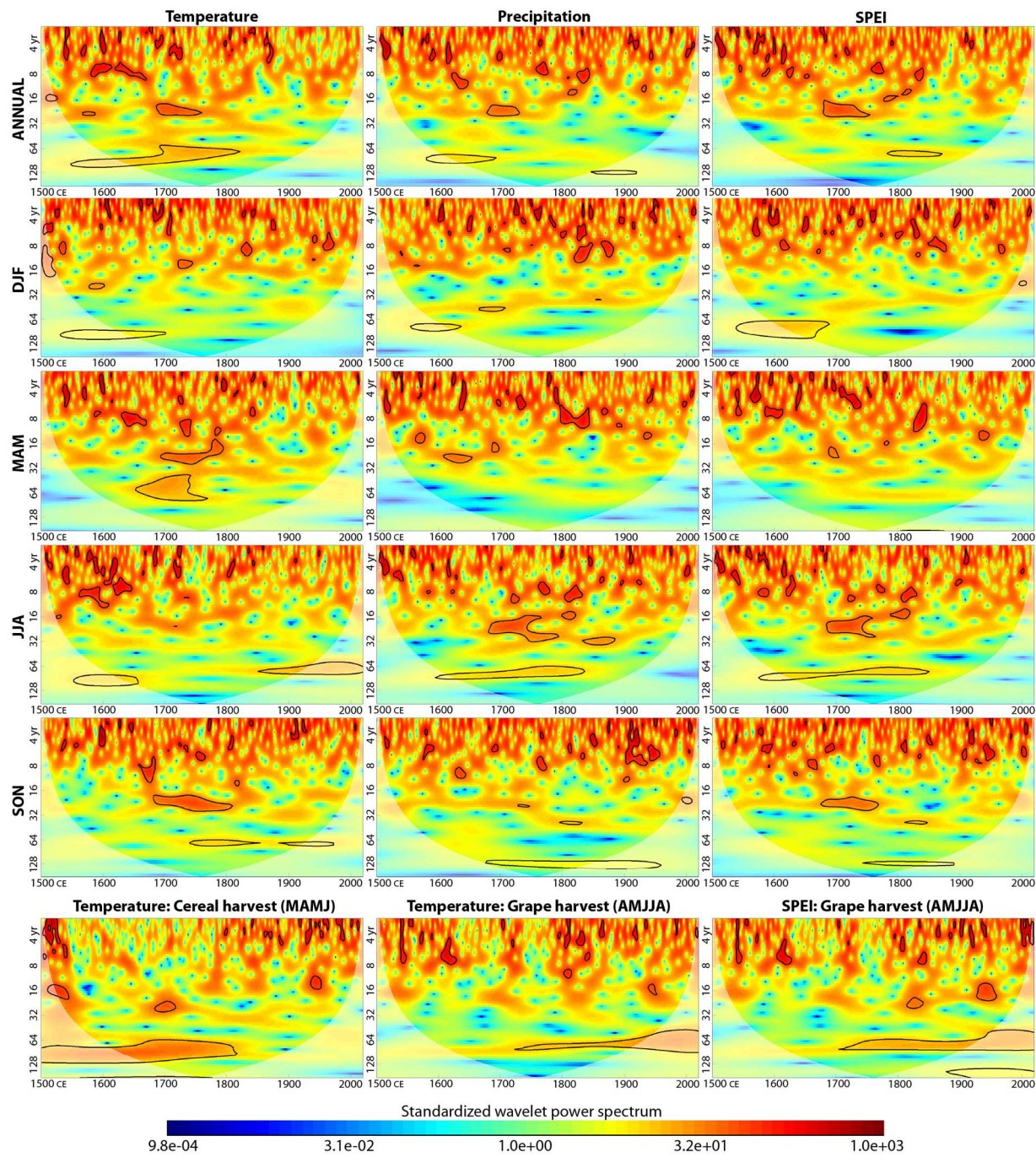

**Figure 9.** Standardized wavelet power spectra for temperature, precipitation and SPEI in the Czech Lands for the 1501–2020 period. Statistical significance is highlighted at the 95% level (black line); series preprocessed by removing the GHGRF-correlated trend component.

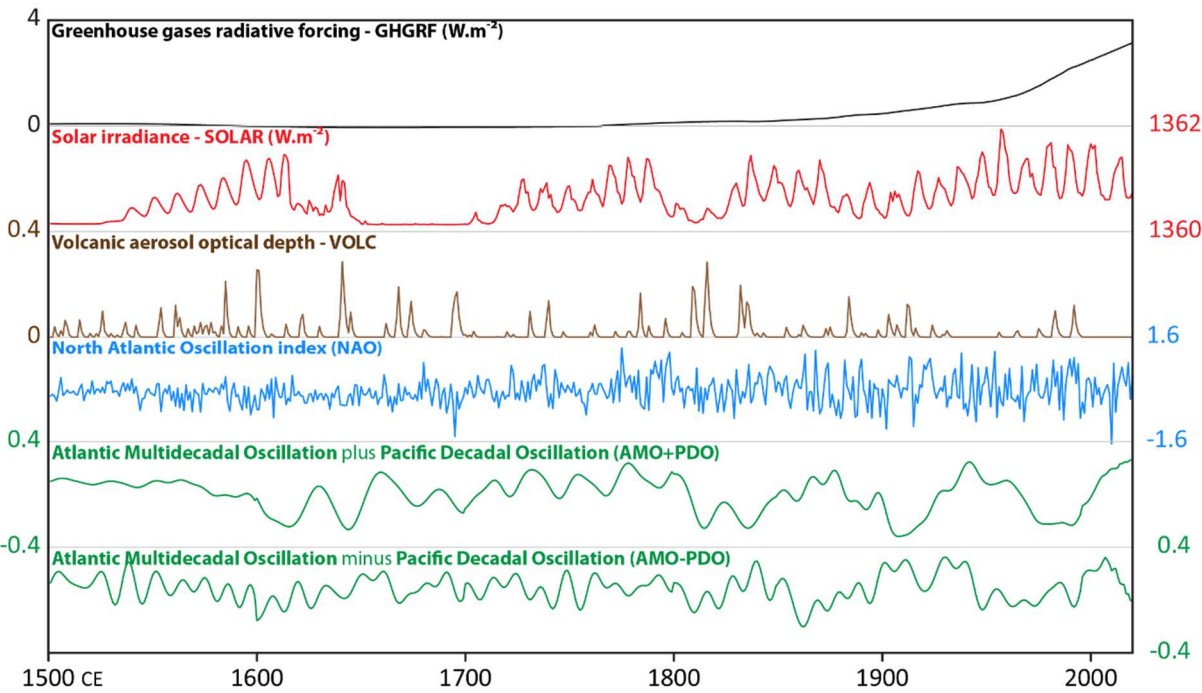

**Figure 10.** Variability in annual series characterizing external forcings and large-scale internal climate oscillations, involved in the attribution analysis.

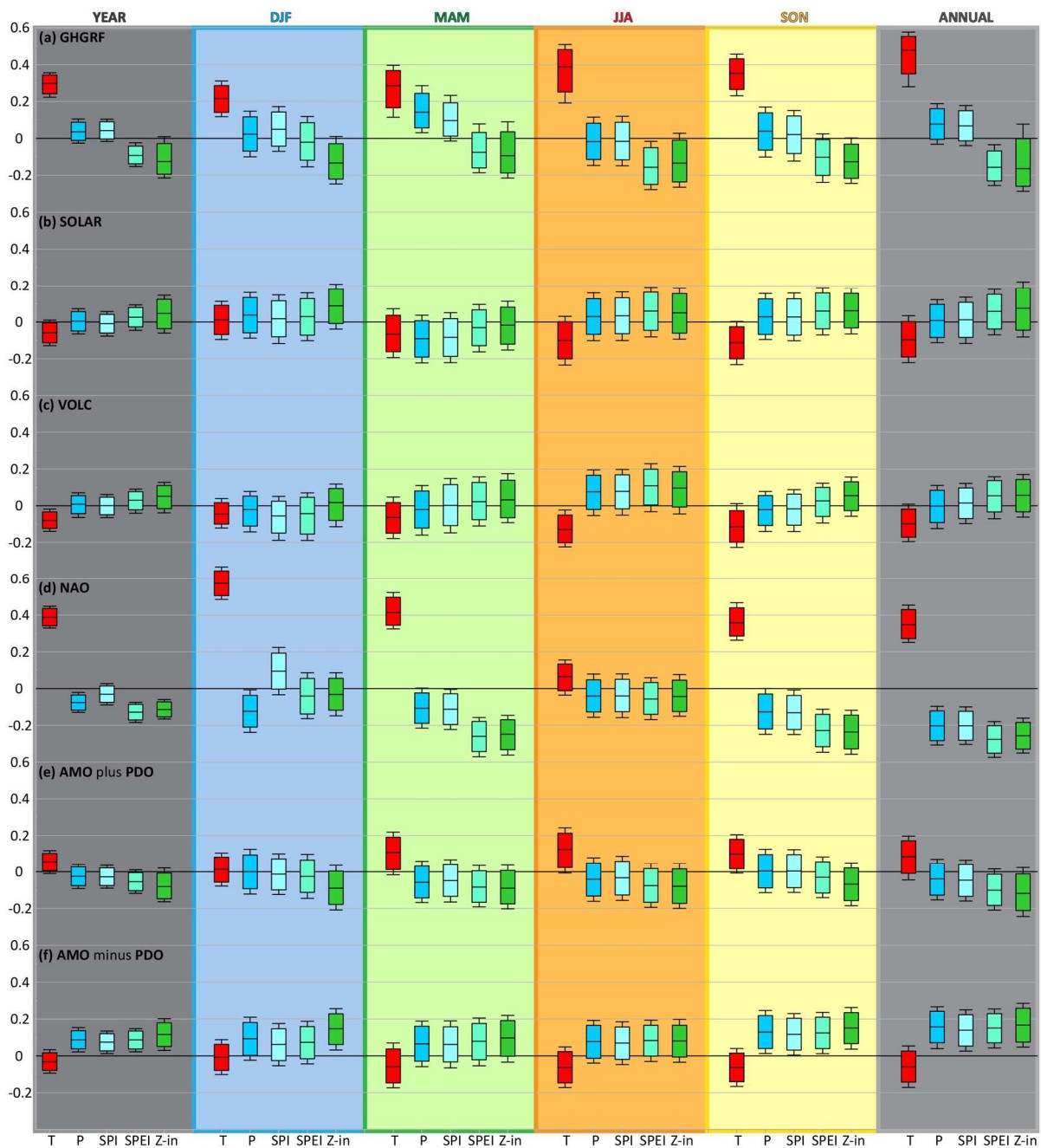

**Figure 11.** Standardized regression coefficients between individual target and explanatory variables and their 95% (box) and 99% (whiskers) confidence intervals. The results shown for time series in seasonal time steps involving all seasons (YEAR), individual seasons analysed separately (DJF, MAM, JJA, SON), and annual averages (ANNUAL).

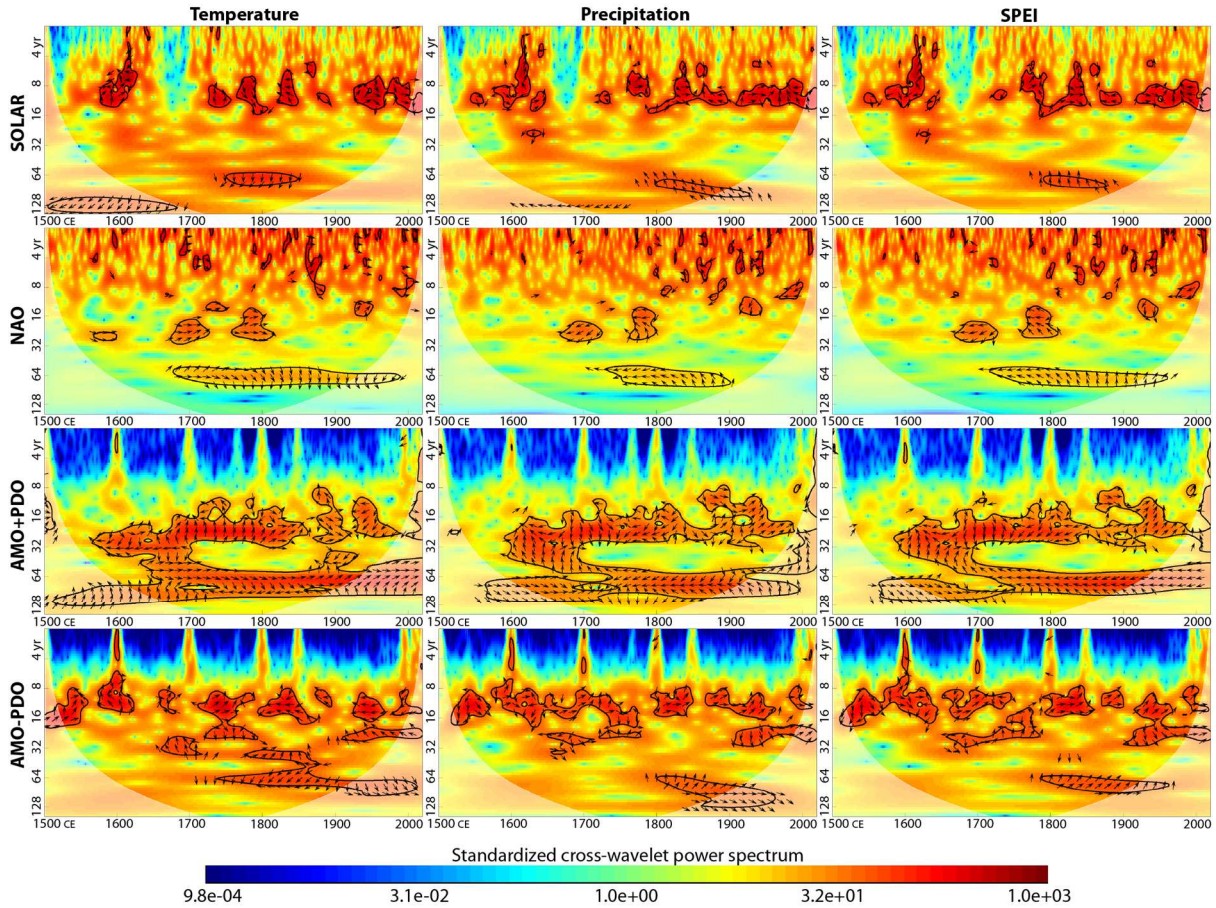

**Figure 12.** Standardized cross-wavelet spectra between series of temperature, precipitation or SPEI and explanatory variables with prominent oscillatory components (all seasons). Arrows show local phase shifts of the two series (with right-facing arrows corresponding to identical phases); areas with statistically significant oscillations are enclosed by black lines (95% confidence level, AR(1) process null hypothesis).

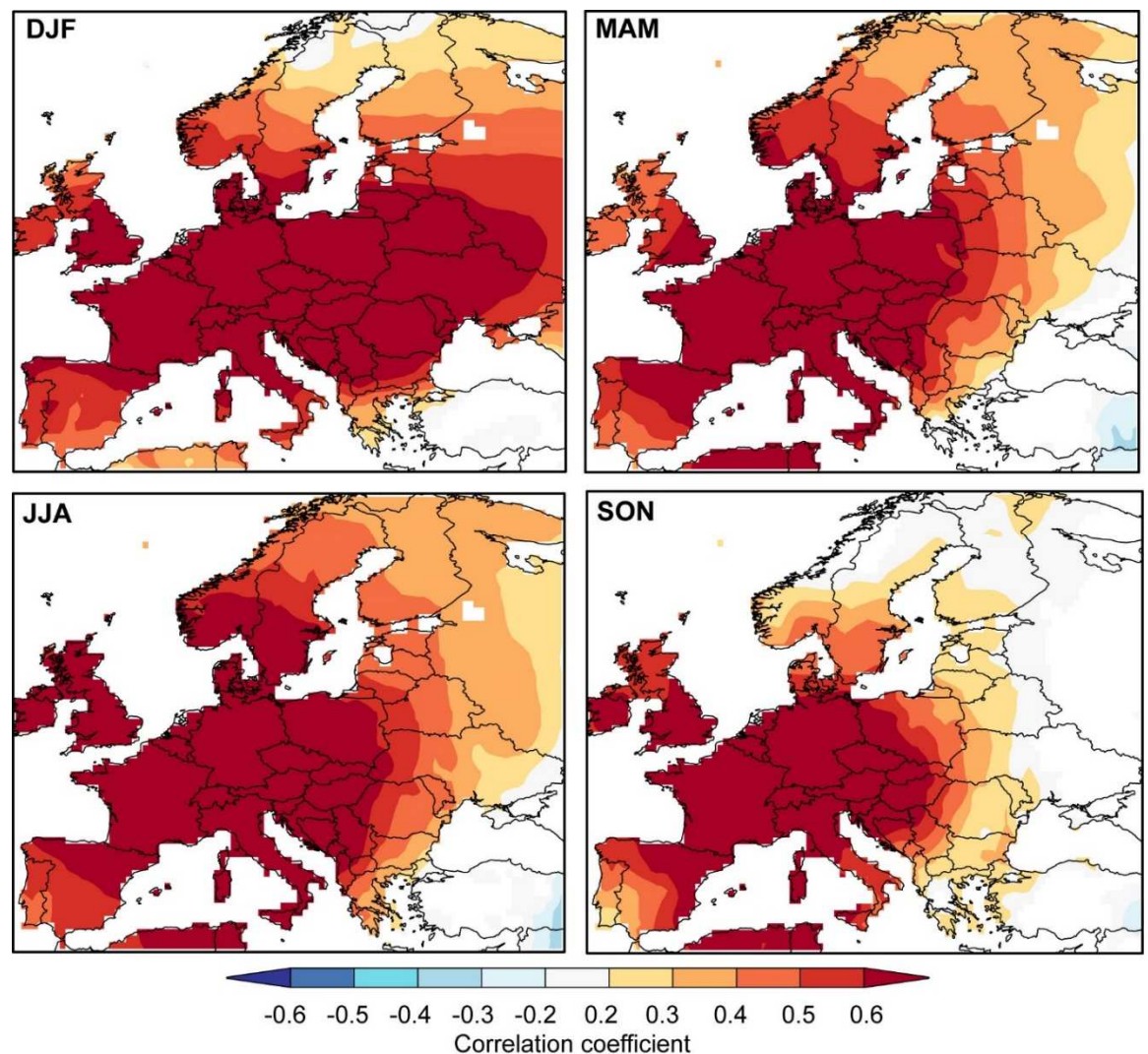

**Figure 13.** Spatial correlations between reconstructed seasonal central European temperature series by Dobrovolný et al. (2010) and gridded European temperature reconstruction by Luterbacher et al. (2004) and Xoplaki et al. (2005) in the 1501–2002 CE period.

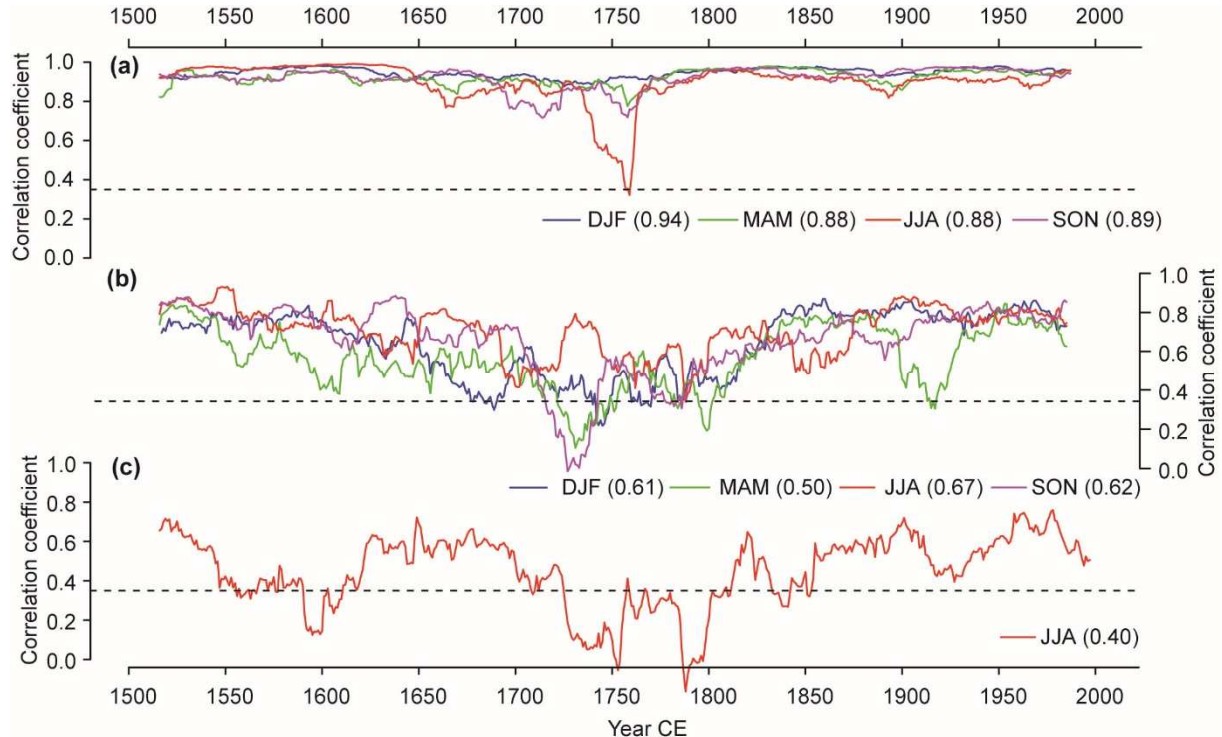

**Figure 14.** Running 31-year correlation coefficients between the seasonal Czech climate reconstructions and selected gridded reconstructions averaged over central Europe (45°N–54°N, 5°E–23°E): (a) central European temperatures (Dobrovolný et al., 2010) vs. temperatures according to Luterbacher et al. (2004) and Xoplaki et al. (2005) for the 1501–2002 period; (b) Czech precipitation (Dobrovolný et al., 2015) vs. precipitation totals according to Pauling et al. (2006) for the 1501–2000 period; (c) Czech JJA scPDSI (Brázdil et al., 2016) vs. JJA scPDSI according to Cook et al. (2015) for the 1501–2012 period. Numbers in brackets represent overall correlation coefficients for the entire common period in question.

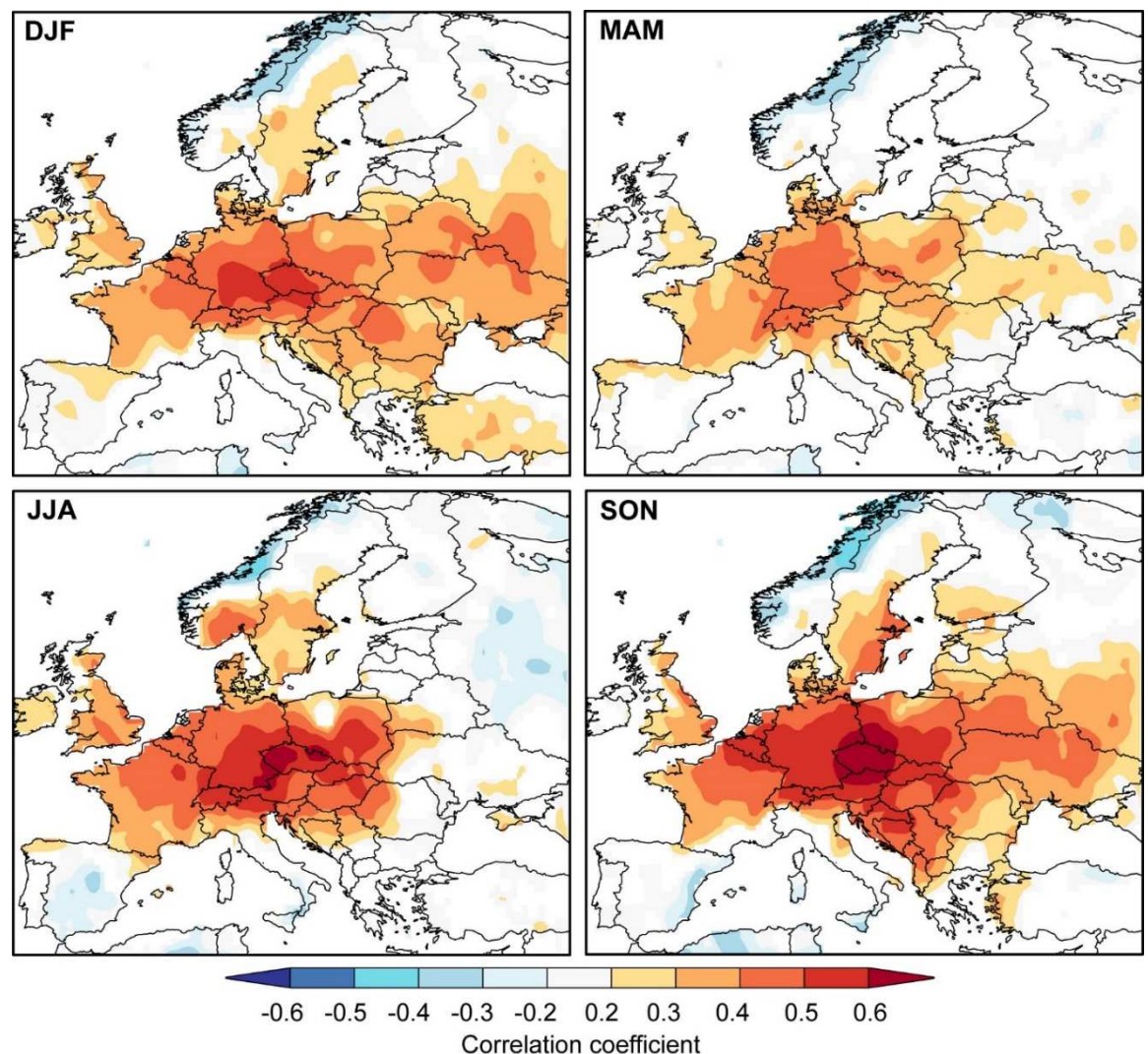

**Figure 15.** Spatial correlations between reconstructed seasonal Czech precipitation series (Dobrovolný et al., 2015) and European gridded precipitation reconstruction (Pauling et al., 2006) for the 1501–2000 CE period.

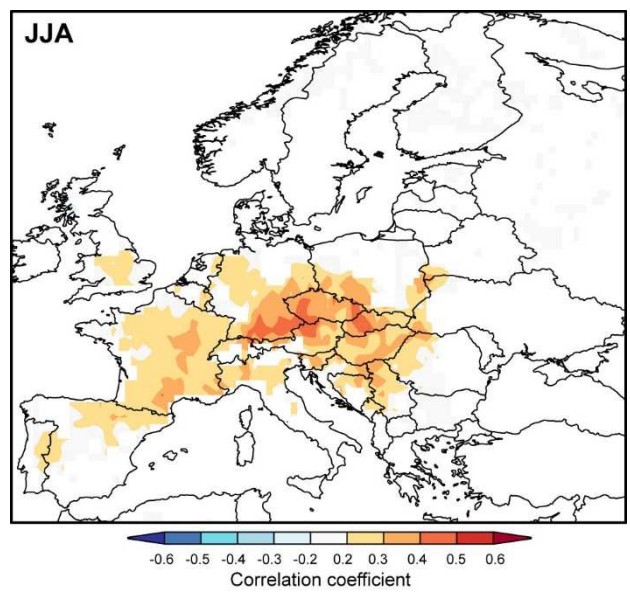

**Figure 16.** Spatial correlations between reconstructed Czech JJA scPDSI series (Brázdil et al., 2016) and gridded European JJA scPDSI reconstruction (Cook et al., 2015) for the 1501–2012 CE period.

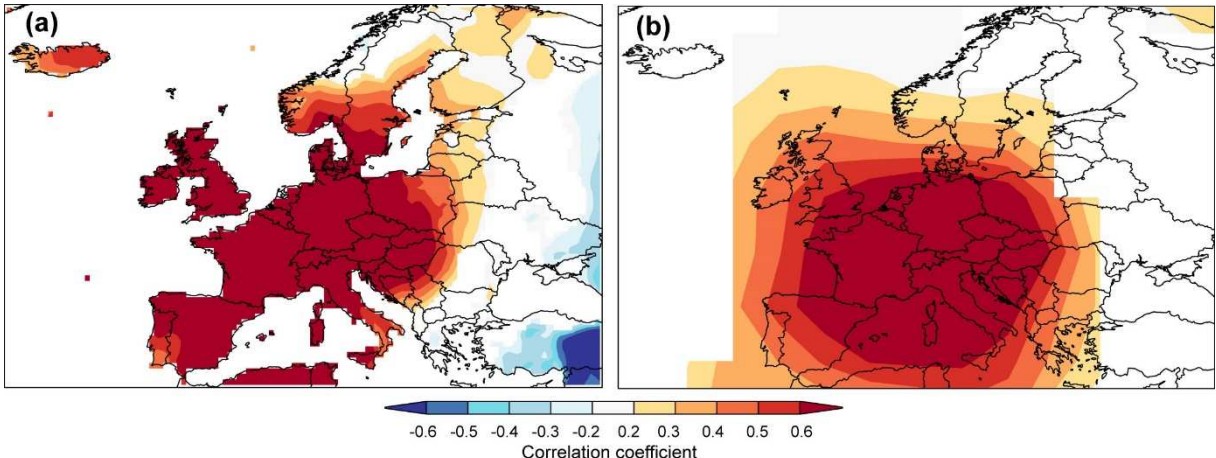

**Figure 17.** Spatial correlations between (a) JJA reconstructed temperatures (Dobrovolný et al., 2010) and temperature field reconstruction (Luterbacher et al., 2004) for the 1600–1750 period; (b) JJA measured temperatures (Dobrovolný et al., 2010) and HadCRUT5.0 temperature field (Morice et al., 2020) for the 1851–2000 period.