# Peer review of "Documentary-based climate reconstructions in the Czech Lands 1501–2020 CE and their European context"

_Climate of the Past, 2021_

## Author Response (AR1)

**Referee #1 – Heinz Wanner**

**General comments**

Based on documentary-based sources, annual and seasonal temperature, precipitation and drought indices were reconstructed in the Czech lands from 1501 to 2020 AD. The study was supplemented by wavelet analyses and an attribution analysis. The temperature series exhibits a statistically significant increasing trend, rising from about 1890 and particularly from the 1970s. In particular, it could be shown that temperature drops in summer are influenced by volcanic events, and that the fingerprint of the North Atlantic Oscillation becomes visible in the other seasons. Certain drought indices show an astonishing decrease over the last decades.

The resulting data set is extremely rich and extensive. The number and scope of the statistical analyses are, in my view very large (e.g. the high number of wavelets), and dynamic analyses are rather sparse. The text is very dense and precisely written, but it is a little short in view of the large number of figures. However, I would rather reduce the number of figures than vote for a text expansion.

I propose to accept the paper after a number of specific revisions.
RESPONSE: We would like to thank Heinz Wanner for a careful evaluation of our paper and raising important critical comments which we are trying to answer below.

**Specific comments**

-Page 3, line 19-24: Is it really necessary to calculate four drought indices? What is the increase in knowledge if the SPEI and the Z-index are added to the SPI and PDSI?
RESPONSE: The four drought indices belong to those used most frequently in drought papers. Each of them shows different aspect of drought both in terms of considered drivers as well as time scale. SPI reflects particularly to the deficit of precipitation compared to normal patterns, SPEI combines effects of precipitation and temperatures including evapotranspiration, Z-index and PDSI reflect particularly soil drought, calculated without memory in monthly step (Z-index) or taking memory of drought into account (PDSI). There is not surprising high relationship between precipitation and SPI, but we do not see it as a reason to exclude SPI from our analysis. Because of reflecting of different aspects of drought, we would like to preserve all four drought indices in our paper since it would make the study useful to wider audience.

-Page 4, line 19-21: Why did you not use the most complete and modern volcanic data, e.g. by Toohey and Sigl, 2017?
RESPONSE: Using Toohey and Sigl (2017) data (eVolv2k) would also be potentially possible, but their dataset only covers period up to 1900 CE (and extension by a different series would therefore be needed). Moreover, as discussed by Toohey and Sigl themselves, only relatively minor differences exist between eVolv2k and prior reconstructions (including volcanic aerosol optical depths by Crowley and Unterman, 2013, i.e. the data employed in our paper) after *c.* 1250 CE, i.e. no major change in volcanism-related results should result from switching to eVolv2k data.

-Page 4, line 28: You suggest to include PDO, combined with AMO. Are you convinced PDO (combined with an AMO Index) can significantly affect the climate of the Czech

Lands? AMO correlates with the NAO and is – in a new paper - additionally questioned as an explaining mode by Mike Mann.

RESPONSE: Regarding inclusion of PDO: as previous analyses (such as Mikšovský et al., 2019) have suggested, there is a quite distinct (and statistically significant) component in multicentennial central European drought series correlated with PDO phase, both on its own and in combination with AMO. This is also reflected in our results (as seen from the regression coefficients in Fig. 11, which indicate a significant link between all the drought indices and the AMO-PDO predictor).

Regarding relation of NAO and AMO: While there certainly may be dynamical links between AMO/AMOC and NAO (a matter that is still a subject of ongoing research and debate), please note that for predictors included in our analysis, almost no correlations exist (as seen from Fig. 10b – now Fig. S1 in the Supplement of the revised manuscript, Pearson correlations of NAO to AMO+PDO and AMO-PDO series are 0.00 and 0.01, respectively). As such, these series each represent a relevant explanatory factor, while being mutually independent (at least in linear statistical sense).

-Page 5, line 39, Fig. 2 a: Can you explain the changing correlations around 1900?

RESPONSE: Accepted, we created the new section 5.1, where we added the paragraph with this explanations (please check it in the context of the whole Section 5.1): "An interesting aspect of lost common signal manifested by a decrease in running correlations below the 0.05 significance level can also appear in the "instrumental part" of the reconstructed series as documented in Fig. 2a. Running correlations of annual temperatures with other five climate variables are highly significant from the 16th century up to the early 19th century. These negative correlations are physically consistent as they show that higher temperatures usually correspond to low precipitation and *vice versa*. Approximately from the mid-19th to the mid-20th centuries correlations among all compared series are not significant. Despite the fact, that annual means express some mixture of different seasonal patterns, this gradual loss of common signal may be interpreted as follows. The fact, that before the 19th century the series are reconstructed from dependent (and thus less variable) temperature and precipitation indices, can be reflected in significant correlations. The instrumental parts of series (target data) are mutually less dependent and more variable than indices. The same patterns as in annual values (Fig. 2a) are well expressed also in SON series and partly in MAM and JJA series, while they do not occur in DJF series (non-significant correlations over the whole period) (not shown). The stronger common signal (significant negative correlation) occurring during the last decades can be attributed to a clearly expressed opposite tendency of rising temperatures and decreasing drought indices. The same pattern does not change even when correlating the detrended series or when changing the length of the window, for which running correlations were calculated."

-Page 6, line 13 and 14: Can you explain the dryness between 1991 and 2020? The positive temperature trend should nevertheless lead to an increase in humidity and precipitation.

RESPONSE: The expectation that "the positive temperature trend should nevertheless lead to an increase in humidity and precipitation" is not followed by measured data. Despite there is statistically significant and quite dramatic increase in temperatures (cf. Zahradníček et al., 2021), it is not followed by precipitation totals, which are generally keeping the same level without any statistically significant trends (cf. Brázdil et al., 2021). It is then reflected in quite dramatic increase in dryness.

-Page 6 + 7, Figs. 7 and 8: I think the inclusion of phenological data is really excellent!

RESPONSE: Thank you.

-Page 7, Figure 9: For me this Figure looks a little like an "overkill". What is the dynamic interpretation behind the very dense Figures?

RESPONSE: Fig. 9 is meant to illustrate variations of wavelet spectra between different variables and seasons (both their similarities and contrasts), plus to compare the spectral structure of documentary/instrumental series to their phenoclimatic counterparts. For this reason, we decided to include all seasons and a reduced selection of target variables (temperature, precipitation and SPEI). Although this admittedly results in a somewhat sizeable figure, it allows the reader to assess robustness of individual spectral features (or lack thereof). We do not provide a dynamical interpretation specifically for the (cross-)wavelet spectra, as they only consider harmonic oscillations in the data (which are typically not dominant components in the series analysed, and thus only capture part of eventual links); we do however use these results in our aggregate interpretation of the results in Discussion.

-Figure 10, attribution analysis: The information on this Figure is extremely dense and not easily readable. Would it not make sense to simplify the Figure and to sort out the really significant correlations, which can point to significant dynamic processes?

RESPONSE: Fig. 10 may have indeed conveyed information that is not essential to the message of the paper. We have therefore moved the correlation matrix (Fig. 10b) to the Supplement (while the mutual correlations of predictors and predictands may be of some interest to the readers, they have mostly been included to illustrate structure of the regression design matrices). As for correlations pointing to significant dynamic processes, please note that even significant correlations do not necessarily imply dynamical/causal links (e.g., the strongest inter-predictor correlation ($r = 0.45$) is indicated between greenhouse gases forcing and solar activity in our analysis, yet this does not represent an actual causal link). We do therefore not attempt to interpret correlations this way.

-Figures 12 and 13: Same comment as for Fig. 9. Do the numerous figures allow plausible dynamic statements?

RESPONSE: Similarly to Fig. 9, these represent a selection that is supposed to capture differences/similarities between spectra pertaining to different pair-wise relationships (so that the most robust features can be inferred), but only using the most relevant plots (since there are dozens of potential combinations of predictor/predictand/season). Again, the results are not discussed on their own, but rather alongside other analyses in the Discussion. Moreover, we decided to move Fig. 13 to the Supplement (as Fig. S2).

-The question of the spatiotemporal representativeness of the Czech data is extremely important. I only wonder whether 5 Figures are needed for this (Fig. 14 - 18). Figure 15 in particular is highly interesting and should be interpreted further.

RESPONSE: All Figs. 14-18 (newly Figs. 13-17) we see as very important to demonstrated the spatial representativeness with respect to temperatures, precipitation and drought. Moreover, Fig. 18 (newly Fig. 17) shows if this spatial representativeness depends on reconstructed (from documentary data) and measured parts of our 520-year series (the related paragraph was moved to the end of Section 4.4, where it fits better than in Discussion). All these figures we see as very important in the manuscript to show European context of our Czech series. To follow the referee request we tried to extend description to Fig. 15 (newly Fig. 14) in different parts of the new Section 5.1 (please check in the context of the whole new section): "However, a closer look at relationships between the two compared reconstructions in Figure 14a reveals another problem. Calculation of JJA temperature differences between reconstructions by Dobrovolný et al. (2010) and Luterbacher et al. (2004)

shows positive differences before the mid-18th century and negative afterward. This shift is responsible for a sharp decrease in running correlations. In order to evaluate this inconsistency, differences of these two series with regard to completely independent JJA multiproxy temperature reconstruction for the Alps by Trachsel et al. (2012) were calculated. For better comparison, the series were first transformed to have a mean of zero and a standard deviation of one. While the differences with the series by Dobrovolný et al. (2010) were distributed more or less randomly around zero, the differences with the Luterbacher et al. (2004) series showed the same patterns as described above: positive differences before the 1750s (i.e., higher temperatures by Trachsel et al., 2012) and negative differences afterward. This indicates that the problem of lost coherence around the 1750s in Fig. 14a cannot be attributed to Dobrovolný et al. (2010) reconstruction."

**Formal aspect**
Reconsider the order of quotations with the same name: Oldest or youngest quotation first?
RESPONSE: We used standard style of quotations as requested by the journal.

**Anonymous Referee #2**

The paper is interesting in that it (i) gives a synthesis of weather and climate changes in the Czech Republic in the period 1501–2020 based on documentary evidence and instrumental observations, (ii) tries to describe the main causes of climate change in this time using statistical attribution analysis (regression and wavelet techniques), and finally (iii) investigates spatiotemporal relationships with gridded European climate reconstructions. All three of these topics are very important for scientists interested in historical climate reconstructions, and especially in those based on documentary evidence.

To be published in the journal, however, the paper needs some substantial improvements and corrections, propositions for which are listed below:

RESPONSE: We thank the reviewer for careful evaluation of our paper and rising critical comments we are trying respond below.

Major weaknesses:
1. In many places the paper has too much of a descriptive character. For example, page 6, lines 4–21. It is very difficult for the reader to follow the text and even more difficult to identify the main findings.

I suggest making a Table showing warmest, coldest, wettest and driest 30-year periods, or maybe even the three warmest, coldest, etc. periods for all indices.

RESPONSE: Accepted. We supposed that it is not necessary to repeat information, which appears already at box-plots in the corresponding figures and again in the text. But to follow the reviewer request, we added related new table as follows:

Table 1. The warmest and driest (a) and the coldest and wettest (b) 30-year periods in annual and seasonal series of climate variables (CV) in the Czech Lands in 1501–2020 CE: T – temperature, P – precipitation, SPI, SPEI, Z-in (Z-index) and PDSI – drought indices

(a) Warmest (T) and driest (P, SPI, SPEI, Z-in, PDSI)

| CV | Annual | DJF | MAM | JJA | SON |
|------|-----------|-----------|-----------|-----------|-----------|
| T | 1991–2020 | 1988–2017 | 1991–2020 | 1991–2020 | 1991–2020 |
| P | 1699–1728 | 1725–1754 | 1773–1802 | 1700–1729 | 1605–1634 |
| SPI | 1704–1733 | 1680–1709 | 1773–1802 | 1700–1729 | 1605–1634 |
| SPEI | 1990–2019 | 1680–1709 | 1989–2018 | 1990–2019 | 1605–1634 |
| Z-in | 1990–2019 | 1991–2020 | 1991–2020 | 1990–2019 | 1990–2019 |
| PDSI | 1991–2020 | 1991–2020 | 1991–2020 | 1991–2020 | 1991–2020 |

(b) Coldest (T) and wettest (P, SPI, SPEI, Z-in, PDSI)

| CV | Annual | DJF | MAM | JJA | SON |
|------|-----------|-----------|-----------|-----------|-----------|
| T | 1829–1858 | 1572–1601 | 1832–1861 | 1569–1598 | 1757–1786 |
| P | 1912–1941 | 1555–1584 | 1885–1914 | 1568–1597 | 1910–1939 |
| SPI | 1912–1941 | 1555–1584 | 1894–1923 | 1568–1597 | 1910–1939 |
| SPEI | 1569–1598 | 1555–1584 | 1873–1902 | 1569–1598 | 1910–1939 |
| Z-in | 1912–1941 | 1898–1927 | 1876–1905 | 1569–1598 | 1887–1916 |
| PDSI | 1913–1942 | 1913–1942 | 1888–1917 | 1913–1942 | 1912–1941 |

2. I suggest taking into account other additional NAO reconstructions: for winter, for example, it is possible to use the index recently proposed by Cook (Cook E. R., D'arrigo R. D., Mann M. E., et al., 2002, A Well-Verified, Multiproxy Reconstruction of the Winter North Atlantic Oscillation Index since A.D. 1400, J. of Climate, Vol. 15,

1754 – 1764, Cook E.R., 2003, Multi-Proxy Reconstructions of the North Atlantic Oscillation (NAO) Index, A Critical Review and a New Well-Verified Winter NAO Index Reconstruction Back to AD 1400. In The North Atlantic Oscillation, Hurrell JW, Kushnir Y, Ottersen G, Visbeck M (eds)).

RESPONSE: It is indeed true that use of a different version of a predictor can alter the outcomes of the attribution analysis (particularly in cases such as ours, when reconstructed data are used in the roles of both target and explanatory variables). Note, however, that effects of using alternative NAO reconstructions were already examined in our prior analysis (Mikšovský et al, 2019), utilizing a similar test setup and using NAO data by Trouet et al. (2009, doi 10.1126/science.1166349) and Ortega et al. (2015, doi 10.1038/nature14518), in addition to the Luterbacher et al. (2001) series. Luterbacher et al. (2001) data were found to have the generally strongest correlation with Czech climate reconstructions (and the respective links were found to be quite stable, throughout the entire five-century span of the data). We therefore opted for use of Luterbacher et al. (2001) NAO series in the current paper. Additionally, in the specific case of Cook et al. (2002) reconstruction, suggested by the reviewer, its winter-only character would not allow for our analysis to be carried out in its intended all-season scope, so we would prefer to not use it in our current paper.

3. Generally, all four drought indices are well correlated (Table 1), and I therefore suggest limiting their number to two indices. The text describing the results will be more concise and readable. The best choice in my view is to use SPI and SPEI. SPEI is the index best correlated with temperature and precipitation in all seasons, and, moreover, only this index was independently reconstructed for the Czech Republic using phenological data.

RESPONSE: The four drought indices belong to those used most frequently in papers analysing droughts. Each of them shows different aspect of drought both in terms of considered drivers as well as time scale. SPI reflects particularly to the deficit of precipitation compared to normal patterns, SPEI combines effects of precipitation and temperatures including evapotranspiration, Z-index and PDSI reflect particularly soil drought, calculated without memory in monthly step (Z-index) or taking memory of drought into account (PDSI). Because PDSI is the most complex and broadly used index for drought evaluation (for example, PDSI is used in dendroclimatological reconstructions), we would like to preserve both PDSI (including drought memory) and Z-index, expressing drought without such drought memory (similarly as SPI and SPEI). Furthermore, despite correlations calculated between climate variables for the whole series being high in some cases, their partial components may behave very different (for example, the trend correlated with GHGRF in DJF is different for SPEI and for Z-index, including differences in statistical significance – see Fig. 11). SPEI calculated from phenological data we count less representative than SPEI calculated from temperature and precipitation indices.

4. In the Discussion section a comparison of the obtained results against other similar climate reconstructions of local and regional character available for the central and other parts of Europe should be also presented.

RESPONSE: Accepted. To follow the reviewer comments, we created a new section 5.1, in which the following paragraphs are particularly relevant to addressing this comment (please check in the context of the whole section):

[revised manuscript text omitted]

5. The attribution analysis must be done separately – for pre-instrumental (reconstructed series) and instrumental periods at least. For example, for the periods 1501–1800(50) and 1801(51)–2020. It is obvious that until about the mid-19th century climate changes were caused mainly by naturals factors (volcanic and solar forcing). Anthropogenic factors (mainly greenhouse gases) are important only for the industrial period and therefore should be limited to this period.

RESPONSE: Please note that such application of regression analysis to shorter data segments was already carried out in a prior paper, Mikšovský et al. (2019), where sub-periods 1501-1850 and 1851-2006 were considered separately in addition to the full length of the series. We did not deem it useful to repeat these partial tests in the current paper, as the conclusion would likely be near-identical to those in Mikšovský et al. (2019). Furthermore, using shorter data segments (and thus fewer data points) increases the uncertainty of the regression coefficients (i.e., the size of the respective confidence intervals), making the attribution analysis less sensitive. This even applies to the analysis of long-term trends such as those related to greenhouse gases forcing – even when the predictor only exhibits noteworthy variability in a part of the analysis period, using the entire length of available data allows the regression mapping to better quantify the link to target variable(s), and to more reliably distinguish between different sources of trend-like changes.

Minor weaknesses:
1. 5, line 39 – please explain the reason for such a big change in correlation coefficients (from about Ë   0.7 to 0.0–0.2, Fig. 2a) around 1900 between all studied series. What happened at the end of the 19th century and the beginning of the 20th century that the correlation between temperature and other variables was lost? Is this a problem of loss of homogeneity of temperature or precipitations series?

RESPONSE: Accepted. Response to this comments is included in the following paragraph in the newly created section 5.1 (please check in the context of the whole section):

"An interesting aspect of lost common signal manifested by a decrease in running correlations below the 0.05 significance level can also appear in the "instrumental part" of the reconstructed series as documented in Fig. 2a. Running correlations of annual temperatures with other five climate variables are highly significant from the 16th century up to the early 19th century. These negative correlations are physically consistent as they show that higher temperatures usually correspond to low precipitation and *vice versa*. Approximately from the mid-19th to the mid-20th centuries correlations among all compared series are not significant. Despite the fact, that annual means express some mixture of different seasonal patterns, this gradual loss of common signal may be interpreted as follows. The fact, that before the 19th century the series are reconstructed from dependent (and thus less variable) temperature and precipitation indices, can be reflected in significant correlations. The instrumental parts of series (target data) are mutually less dependent and more variable than indices. The same patterns as in annual values (Fig. 2a) are well expressed also in SON series and partly in MAM and JJA series, while they do not occur in DJF series (non-significant correlations over the whole period) (not shown). The stronger common signal (significant negative correlation) occurring during the last decades can be attributed to a clearly expressed opposite tendency of rising temperatures and decreasing drought indices. The same pattern does not change even when correlating the detrended series or when changing the length of the window, for which running correlations were calculated."

2. 8a – a similar problem to that mentioned in point 1: please explain the reasons for the loss of correlations between the two reconstructed temperature series only just after the mid-17[th] century and mid-18[th] century for two–three decades.

RESPONSE: Accepted. We tried to explain this problem and general loss of coherence among different reconstructions in the newly created section 5.1, where we reported also weaknesses in both "phenologically-based" reconstructions (please check it in the context of the whole new section). Particularly the following paragraphs concern of the above problem:

"As for series derived from phenological data, MAMJ temperatures reconstructed from winter wheat harvest dates were compared with 11 late spring and summer temperature series in central Europe (see Fig. 6 in Možný et al., 2012). Better coherence was found with documentary-based and biophysically-based reconstructions (harvest dates) than those based on tree-rings. A significant drop in correlations appeared particularly in the second half of the 17th century and around the 1750s. This may be partly related to the problem in the data quality of the winter wheat harvest dates. These dates had to be recalculated from the harvest dates of other available cereals in periods when the winter wheat dates were not available. The distinct role may be attributed to the "social bias" in data related to the complicated social and political situation in the country (see discussion related to those periods in Možný et al., 2012, and also Fig. 8a in the current study).

Similarly, AMJJ temperatures reconstructed from grape harvest dates were compared with 17 European temperature reconstructions based on temperature indices derived from documentary data, grape harvest dates, tree-rings, and multiproxies (see Fig. 9 in Možný et al., 2016a). Possible inconsistencies were found in the first half of the 16th century, around 1650, 1750, and 1900. Four periods with potential "social bias" were identified in the last decades of the 16th century and then in the 1640s–1670s, 1750s–1780s, and 1850s–1910s."

Could you also inform the reader which of the temperature reconstructions presented in Fig. 8a is better and more reliable (based on temperature indices or on wheat harvest dates). Differences in absolute values of temperature are sometimes very large. This is very well seen particularly in the aforementioned times when the correlation is lost.

RESPONSE: We understand the reviewer comment, but the answer will very much depend on the chosen criteria. Each of these reconstructions is based on different type of data with some advantages and disadvantages. For example, if we will take into account the explained variance in the calibration/verification period, both reconstructions are comparable. The wheat harvest day (WHD) reconstruction explains 0.70 of the MAMJ temperatures and it is 0.69 in case of the central European temperature (CEUT) reconstruction (mean value for the corresponding months). From direct comparison in Figure 8a (bottom) it follows that the WHD captures the low frequency signal better than the CEUT. However, this is with a high probability related to the quality of data used for the WHD chronology compilation. The periods that show the largest differences in the two compared reconstructions in Fig. 8a well correspond to a significant drop in correlations. As can be verified from the Figure 6 of Možný et al. (2012) these suspicious periods, especially the second half of the 17th century and the period centred in 1750s, can be well identified when one compares the WHD with several other proxy reconstructions in central European context. This indicates that the problem probably lies in the quality of the data used to compile the WHD chronology that is changing over time. This explanation may be supported by the fact that also the variability of the WHD-based temperatures is clearly changing over time (see Figure 7a, top).

3. 8 – the same scale should be used in Figures 8a and 8b for temperature in both types of reconstruction comparisons, i.e. four degree distance between lowest and highest values.

RESPONSE: Accepted, the new version of figure was prepared as requested.

4. Figs 14 and 16 – for winter you can compare your results with Luterbacher et al. (2010) similar calculations made for Poland area and Europe using also modelling works: Luterbacher J., Xoplaki E., Küttel M., Zorita E., González-Rouco J. F., Jones P. D., Stössel M., Rutishauser T., Wanner H., Wibig J., Przybylak R., 2010, Climate Change in Poland in the Past Centuries and Its Relationship to European Climate: Evidence From Reconstructions and Coupled Climate Models. in: Przybylak R, Majorowicz J, Brázdil R, Kejna M (eds) The Polish Climate in the European Context: An Historical Overview, Springer, Berlin Heidelberg New York, 3-39.

RESPONSE: Trying to follows this comment, we asked for corresponding data the first author of the paper, Prof. Juerg Luterbacher (WMO, Geneva), but he replied that he no longer has any such data. On his recommendation we contacted also one of Polish co-authors, Prof. Rajmund Przybylak (UMK, Torun), but with the same negative result.

5. I suggest reducing the number of figures and presenting more possible explanations for peculiarities in the course of climate change in the Czech Republic in the study period.

RESPONSE: Accepted. To reduce the number of figures in the main manuscript, the wavelet coherence plots (originally in Fig. 13) have been moved to the Supplement, as Fig. S2. Furthermore, in response to a suggestion by reviewer 1, Fig. 10 has been simplified and the correlation matrix (originally Fig. 10b) moved to the Supplement as Fig. S1. Concerning of other figures in the manuscript, we consider every of them as important and we would like to preserve them in the manuscript. We extended manuscript in the parts, where it was requested by both referees (see the new section 5.1 and our responses above), and we believe that we have explained basic peculiarities in the course of climate change in the Czech Republic.

I can recommend acceptance of the manuscript for publication in the *Climate of the Past* only on the condition that the remarks and suggestions listed above are satisfactorily taken into account.

---

## Author Response (AR2)

**Comments to the author – Chantal Camenisch:**

Please add in your article briefly your explanations in your author's response to questions 2 (alternative NAO reconstructions) and 5 (regression analysis to shorter data segments) of the original referee report 2.

**Question 2:**
The requested paragraph was added as the last in Section 5 Discussion:
"Finally, note that the outcomes of the attribution analysis may also be subject to specific properties of the explanatory variables used, particularly in case of the reconstructed indices of internal climate variability modes (NAO, AMO, PDO). This issue has been previously investigated in Mikšovský et al. (2019), where multiple independent reconstructions were used for each of these indices (including NAO reconstructions by Trouet et al., 2009 or Ortega et al., 2015, AMO reconstruction by Gray et al., 2004, and PDO reconstructions by MacDonald and Case, 2005 or Shen et al., 2006). The series based on data by Luterbacher et al. (2001) and Mann et al. (2009) were shown to carry the relatively strongest link to central European climate variability, and were therefore employed in this current analysis. Even so, the problem of predictor-related robustness of the attribution analysis remains an essential one and an issue worthy of revisiting, especially when new relevant proxy-based data arrive in the future."

**Question 5:**
The requested explanation was added to Section 4.3 Attribution analysis at the end of the first paragraph:
"… Note that, unlike in prior analysis presented in Mikšovský et al. (2019), the El Niño – Southern Oscillation (ENSO) was not included among the explanatory factors due to largely negligible influence exhibited by the available ENSO reconstructions covering our target period. We also do not present results obtained separately for instrumental and pre-instrumental periods as was done in Mikšovský et al. (2019), because such division tends to magnify uncertainties pertaining to identification of slow-variable components in climatic time series. Furthermore, outcomes for PDSI are not shown due to the long memory component in this drought index, making proper pairing of predictand and predictors problematic without additional transformations."

All new references were added to their list.